# Nitric oxide modulates contrast suppression in a subset of mouse retinal ganglion cells

Dominic Gonschorek[1,2,3], Matías A Goldin[4], Jonathan Oesterle[1,2,5], Tom Schwerd-Kleine[1,2,3], Ryan Arlinghaus[1,2], Zhijian Zhao[2], Timm Schubert[1,2], Olivier Marre[4], Thomas Euler[1,2,3,6]*

[1]Werner Reichardt Centre for Integrative Neuroscience, University of Tübingen, Tübingen, Germany; [2]Institute for Ophthalmic Research, University of Tübingen, Tübingen, Germany; [3]GRK 2381 'cGMP: From Bedside to Bench', University of Tübingen, Tübingen, Germany; [4]Institut de la Vision, Sorbonne Université, INSERM, CNRS, Paris, France; [5]Hertie Institute for AI in Brain Health, Tübingen AI Center, University of Tübingen, Tübingen, Germany; [6]Bernstein Center for Computational Neuroscience, University of Tübingen, Tübingen, Germany

## eLife Assessment

This **important** study is the first comprehensive analysis of the modulatory effects of nitric oxide (NO) on the response properties of retinal ganglion cells (RGCs) in the mouse retina using two-photon calcium imaging and multi-electrode arrays (MEA). The results provide **compelling** evidence that a subset of suppressed-by-contrast RGCs are affected. These unexpected findings are likely of broad interest to visual neuroscientists.

*For correspondence: thomas.euler@cin.uni-tuebingen.de

**Abstract** Neuromodulators have major influences on the regulation of neural circuit activity across the nervous system. Nitric oxide (NO) has been shown to be a prominent neuromodulator in many circuits and has been extensively studied in the retina. Here, it has been associated with the regulation of light adaptation, gain control, and gap junctional coupling, but its effect on the retinal output, specifically on the different types of retinal ganglion cells (RGCs), is still poorly understood. In this study, we used two-photon $Ca^{2+}$ imaging and multi-electrode array (MEA) recordings to measure light-evoked activity of RGCs in the ganglion cell layer in the ex vivo mouse retina. This approach allowed us to investigate the neuromodulatory effects of NO on a cell type-level. Our findings reveal that NO selectively modulates the suppression of temporal responses in a distinct subset of contrast-suppressed RGC types, increasing their activity without altering the spatial properties of their receptive fields. Given that under photopic conditions, NO release is triggered by quick changes in light levels, we propose that these RGC types signal fast contrast changes to higher visual regions. Remarkably, we found that about one-third of the RGC types, recorded using two-photon $Ca^{2+}$ imaging, exhibited consistent, cell type-specific adaptational response changes throughout an experiment, independent of NO. By employing a sequential-recording paradigm, we could disentangle those additional adaptational response changes from drug-induced modulations. Taken together, our research highlights the selective neuromodulatory effects of NO on RGCs and emphasizes the need of considering non-pharmacological activity changes, like adaptation, in such study designs.

## Introduction

The retina can be considered as a visual signal processor with a straightforward functional architecture (*Masland, 2001*; *Wässle and Boycott, 1991*; *Wässle, 2004*). The photoreceptors convert the stream of photons from the environment into an electrical signal, which is relayed downstream by the bipolar cells (BCs) to the retinal ganglion cells (RGCs), the tissue's output neurons. Along this vertical pathway, the visual signal is shaped by two lateral, inhibitory cell classes: horizontal cells in the outer and amacrine cells (ACs) in the inner retina. The resulting intricate networks allow sophisticated computations, reflected in the >40 output channels that relay diverse visual feature representations as spike trains to higher brain regions (*Baden et al., 2016*; *Goetz et al., 2022*; *Seabrook et al., 2017*).

Over the past decades, the retinal synaptic networks have been intensively studied, which greatly broadened our understanding of early visual signal processing (*Anderson et al., 2011*; *Dunn and Wong, 2014*; *Helmstaedter, 2013*; *Helmstaedter et al., 2013*; *Johnston and Lagnado, 2015*; *Marc et al., 2013*). Specifically, the availability of connectome data for large parts of the retina helped in unprecedented ways, as can be seen from studies, e.g., investigating direction-selective circuits and their precise wiring regarding starburst ACs and direction-selective RGCs (e.g. *Briggman et al., 2011*; *Ding et al., 2016*).

What is not well captured by connectomic approaches are 'wireless' interactions mediated by neuromodulators, which comprise a broad variety of very different small molecules, including monoamines (e.g. dopamine [*Roy and Field, 2019*; *Witkovsky, 2004*], histamine [*Greferath et al., 2009*; *Warwick et al., 2022*], serotonin [*Masson, 2019*]), endocannabinoids (*Schwitzer et al., 2016*; *Yates et al., 2022*; *Yazulla, 2008*), gasotransmitters such as nitric oxide (NO) (*Goldstein et al., 1996*; *Vielma et al., 2012*), as well as a variety of neuropeptides (e.g. neuropeptide Y [*Santos-Carvalho et al., 2014*; *Santos-Carvalho et al., 2015*]). Only few of the >20 neuromodulators (*Yan et al., 2020*) found in the retina are released from centrifugal fibers (e.g. histamine and serotonin [*Gastinger et al., 2006*; *Greferath et al., 2009*; *Masson, 2019*; *Warwick et al., 2022*]), whereas most of them are released in addition to GABA or glycine by ACs (*Diamond, 2017*; *Hirasawa et al., 2012*; *O'Malley and Masland, 1989*; *O'Malley et al., 1992*; *Yan et al., 2020*). Neuromodulators have long been implicated in adapting the retina to different contextual states necessary to robustly perform in a highly dynamic visual environment (e.g. *Flood et al., 2018*; *Goel and Mangel, 2021*; *Herrmann et al., 2011*; *Jacoby et al., 2018*). For instance, dopamine (DA) is released by a distinct type of AC (*Bauer et al., 1980*; *Djamgoz and Wagner, 1992*; *Godley and Wurtman, 1988*; *Pérez-Fernández et al., 2019*), and has been shown to regulate light adaptation (*Flood et al., 2018*; *Nichols et al., 1967*; *Nir et al., 2000*) via several cellular mechanisms, such as modulation of gap junctional coupling (*Goel and Mangel, 2021*; *Jin and Ribelayga, 2016*; *Kothmann et al., 2009*; *Ribelayga et al., 2008*) and intrinsic conductances (*Hayashida et al., 2009*; *Lasater and Dowling, 1985*). More recently, histamine was proposed to shape the retinal code in a top-down modulatory manner related to the animal's arousal state (*Gastinger et al., 2006*; *Warwick et al., 2022*). For several neuromodulators, the retinal release sites and receptors are known and their effects on cellular properties have been described, however, a comprehensive, function-oriented, and cell-type view of these neuromodulators' functional implications and contributions to visual signal processing is only slowly emerging (e.g. *Warwick et al., 2023*; *Warwick et al., 2022*).

One of the better-studied neuromodulators in the retina is NO. Neuronal nitric oxide synthase (nNOS) is considered the dominant enzyme producing NO relevant for retinal signal processing, and – depending on species – was shown to be present in different retinal cell classes (*Blom et al., 2009*; *Blute et al., 2000*; *Blute et al., 1997*; *Bredt et al., 1990*; *Dawson et al., 1991*; *Eldred and Blute, 2005*; *Palamalai et al., 2006*; *Yamamoto et al., 1993a*; *Yamamoto et al., 1993b*). In the mouse retina, nNOS was mainly found in specific AC types (*Blom et al., 2012*; *Giove et al., 2009*; *Haverkamp and Wässle, 2000*; *Jacoby et al., 2018*). A few years ago, *Jacoby et al., 2018*, demonstrated that one of those types (nNOS-2 AC) controls the light-dependent release of NO in the inner retina. NO can function via two main pathways (*Ahern et al., 2002*): (i) it can bind to the NO guanylate cyclase (NO-GC; also referred to as soluble guanylate cyclase) receptor, triggering the production of the second messenger cyclic guanosine monophosphate (cGMP), which binds to downstream targets (*Eldred and Blute, 2005*; *Garthwaite, 2005*; *Sitaramayya, 2002*), and (ii) via S-nitrosylation (cGMP-independent) by directly modifying certain receptor and channel proteins (*Hess et al., 2005*; *Miyagi et al., 2002*).

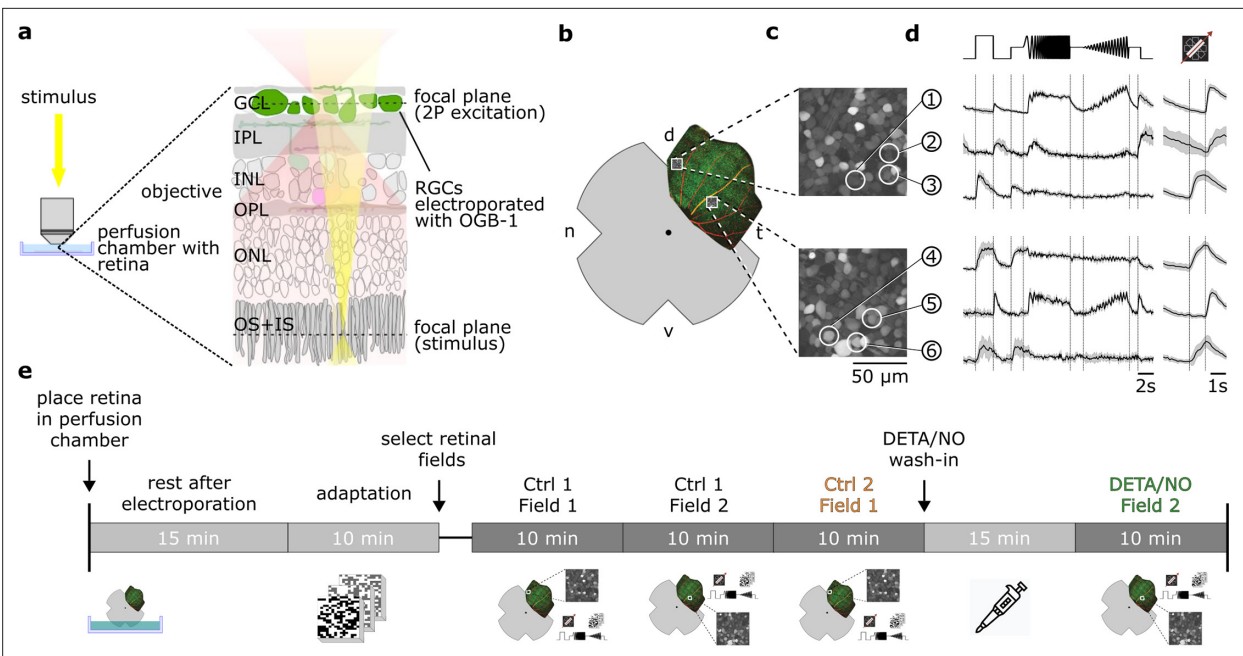

**Figure 1.** Overview of the experimental setup and recording procedure. (**a**) Two-photon imaging of ganglion cell layer (GCL) somata in the whole-mounted ex vivo mouse retina. (**b**) Schematic ex vivo whole-mounted retina (dot marks optic disc; d, dorsal; t, temporal; v, ventral; n, nasal). (**c**) Two representative recording fields from (**b**) showing GCL somata loaded with $Ca^{2+}$ indicator OGB-1 (Methods). (**d**) Representative $Ca^{2+}$ activity from cells in the GCL (white circles in (**c**)) in response to chirp (left) and moving bar stimulus (right) (black, mean; gray, s.d.). (**e**) Timeline of experimental procedure; for details, see text. IPL, inner plexiform layer; INL, inner nuclear layer; OPL, outer plexiform layer; ONL, outer nuclear layer; OS + IS, outer and inner segments; DETA/NO, nitric oxide (NO) donor.

The effects of NO have been primarily linked with light adaptation and the transition between scotopic and photopic signaling pathways via several mechanisms, including uncoupling the gap junctions between AII ACs and On-cone BCs (*Mills and Massey, 1995*), increasing the gain of On-cone BCs in response to dim light (*Nawy and Jahr, 1990*; *Snellman and Nawy, 2004*), and modulating the gain of Off-cone BCs (*Tooker et al., 2013*; *Vielma et al., 2014*). At the retina's output level, increasing NO was found to decrease On- and Off-responses in RGCs (*Wang et al., 2003*). Notably, genetically knocking out nNOS also led to a reduced light sensitivity in RGCs (*Wang et al., 2007*), which was also shown by inhibiting nNOS under light-adapted conditions (*Nemargut and Wang, 2009*). Additionally, NO has been shown to modulate RGC responses via cGMP by altering their cGMP-gated conductances (*Ahmad et al., 1994*; *Kawai and Sterling, 2002*). Taken together, NO can act on different levels and via different pathways in the retina. However, the function of NO at the cell type-level and its role in early visual processing is far from understood.

Here, we systematically studied the functional role of NO and its effects on all retinal output channels in the ex vivo mouse retina. Surprisingly, we observed highly reproducible, cell type-specific changes in the light responses of some RGCs already in the absence of pharmacological manipulation. To account for these adaptational response changes, we developed a recording paradigm to sequentially measure RGC responses under control and drug conditions. We found that NO had a highly selective effect on a distinct set of three RGC types characterized by their suppressed response to temporal contrast, where NO strongly and differentially reduced this suppression, as well as caused them to respond faster. Yet, NO had no discernible effects on their spatial receptive field (sRF) properties. Together, our data suggest that NO modulates the visual feature representation of the retinal output in a highly selective and type-specific manner.

## Results

To investigate NO effects systematically on the retinal output, we performed population imaging from somata in the ganglion cell layer (GCL) of ex vivo mouse retina electroporated with the synthetic

Ca²⁺ indicator Oregon Green BAPTA-1 (*Figure 1a–c*; Methods). In addition, we recorded a complementary dataset using multi-electrode array (MEA) recordings. To identify the different types of RGCs and detect potential NO-induced response changes, we presented a set of visual stimuli, including full-field chirps, moving bars (*Figure 1d*), and binary dense noise (*Baden et al., 2016*) (see Methods).

## A protocol for sequential control/drug recordings

Previous studies have shown that retinal responses recorded with two-photon Ca²⁺ imaging can be systematically affected by experimental factors, such as excitation laser-induced activity, photoreceptor bleaching, and temporal filtering due to Ca²⁺ buffering by the fluorescent indicator (*Euler et al., 2019*; *Euler et al., 2009*; *Zhao et al., 2020*). These changes can be summarized by the umbrella term 'batch effects' (a term coined in the molecular genetics field), which can confound the biological signal and potentially cause erroneous interpretations of the data (*Gonschorek et al., 2021*; *Zhao et al., 2020*). Such batch effects may play a role when, as in our study, data are recorded in a sequential manner to infer possible drug effects.

Because we wanted to detect potentially subtle NO effects, we devised a protocol to make experiments as comparable as possible (*Figure 1e*). After placing the tissue into the recording chamber, it was allowed to recover from the electroporation for 15 min, before we light-adapted the retina for 10 min by presenting the dense noise stimulus. We then selected retinal recording fields, each of which was recorded twice for the complete stimulus set in an interlaced manner (*Figure 1e*). Here, we made sure that the fields were uniformly labeled with the Ca²⁺ indicator and responsive to light stimuli. The first field was recorded twice without perturbation (*control-control*). For the next field, we added the drug to the perfusion medium and incubated the tissue for ~15 min, before recording the field for the second time (*control-drug*). For the last 5–10 min of the wash-in time, we presented the noise stimulus to preserve the light adaptation level. Note that the time between the first and second recording of field 2 was ~15 min longer (the wash-in time) than that of field 1 (see Discussion). For the Ca²⁺ data, we decided against recording also after wash-out, because response quality decreased for the third scan of the same field, likely due to bleaching of fluorescent indicator and photopigment. However, we did include wash-out in the MEA-dataset (see below).

Our sequential-recording protocol yielded paired data at the cell level, allowing us to track if and how each cell's responses changed under the drug condition. Using this protocol, we recorded the following datasets: (i) a control-dataset to test response stability, i.e., Ctrl 1 and Ctrl 2, (ii) a strychnine-dataset to test the protocol for a drug with well-described effects, i.e., Ctrl 1 and Strychnine (1 µM), and (iii) a NO-dataset to infer NO-induced response changes, i.e., Ctrl 1 and NO (DETA/NO; 100 µM). The control-dataset was leveraged to reveal NO-induced effects on the background of potential non-specific response changes throughout the experiment. For the following analyses, we used 3975 RGCs ($n_{Ctrl}$ = 1701; $n_{NO}$ = 1838, $n_{Strychnine}$ = 436) that fulfilled our response quality filtering (see Methods).

## Identifying functional RGC types using a classifier

The mouse retina contains more than 40 RGC types (*Baden et al., 2016*; *Goetz et al., 2022*). As we wanted to investigate if the tested drugs differentially affect the different retinal output channels, we applied an RGC type classifier (*Figure 2a*; *Qiu et al., 2022*), which had been trained and validated on a previously published RGC Ca²⁺ imaging dataset (*Baden et al., 2016*). The classifier predicts a GCL cell's functional type based on soma size and the responses to chirp and moving bar stimuli (see Methods). The classifier also distinguishes between RGCs and displaced ACs. Here, we focused our analysis on the RGCs. To match the conditions under which the classifier's training data was acquired as closely as possible, we predicted types based on the responses from the first control recording (Ctrl 1) and the cells retained their type over the course of their experiment (no re-typing). To minimize classification uncertainty, we additionally discarded cells with low confidence scores (< 0.25, see Methods for details).

We found that the distributions of the predicted RGC types in our datasets matched that of the earlier dataset quite well (*Figure 2b–d*). Also, the predicted mean traces for the individual RGC types were very similar to those in *Baden et al., 2016* (*Figure 2e*), as indicated by the high correlations of their chirp and moving bar responses (*Figure 2f*, left and right, respectively). That the moving bar responses were more strongly correlated than the chirp responses is likely due to the lower complexity

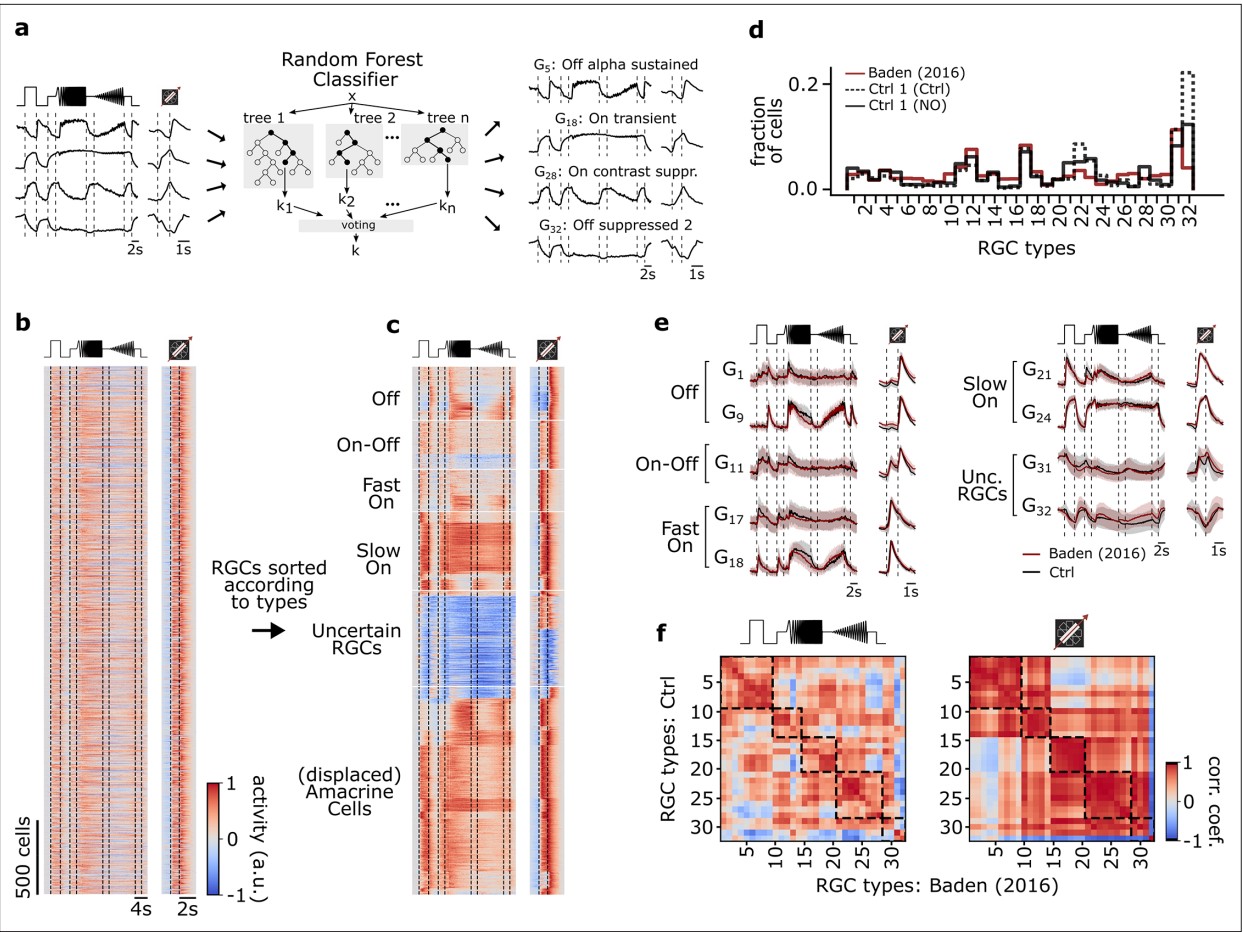

**Figure 2.** Functional classification of mouse retinal ganglion cell (RGC) types. (**a**) Illustration of the random forest classifier (RFC) to predict cell-type labels for Ctrl 1 of both datasets. For each cell, Ca$^{2+}$ responses to chirp and moving bar, soma size, and p-value of permutation test for direction selectivity (left) constitute the input to the RFC (center) to predict a cell-type label, i.e., a type G$_X$ (right). For details, see Methods and **Qiu et al., 2022**. (**b**) Pooled heat map of unsorted cell responses to chirp and moving bar stimulus from both Ctrl 1 datasets after quality filtering (QI$_{MB}$>0.6 or QI$_{chirp}$>0.45, and classifier confidence score ≥ 0.25). The color bar indicates normalized response activity. (**c**) Heat map from (**b**), but sorted according to their assigned type. (**d**) Distribution of RGC types predicted by the RFC classifier for both Ctrl 1 of the control- (Ctrl 1 (Ctrl); solid black), of DETA/NO-dataset (Ctrl 1 (NO); dotted black), and for the dataset from **Baden et al., 2016** (red). (**e**) Representative RGC type response averages to chirp and moving bar (Ctrl, black; training dataset, red). (**f**) Correlation matrix of type mean responses per RGC type between Ctrl and **Baden et al., 2016**, dataset for chirp (left) and moving bar (right). Dashed boxes indicate functional groups (Off, On-Off, Fast On, Slow On, and Uncertain RGCs; see **Baden et al., 2016**). The color bar indicates the Pearson correlation coefficient.

The online version of this article includes the following figure supplement(s) for figure 2:

**Figure supplement 1.** Autocorrelation matrix of **Baden et al., 2016**, dataset.

and shorter duration of the former stimulus. Nonetheless, we found the RGC classification overall to be robust and comparable to the original dataset (**Figure 2—figure supplement 1a and b**).

## Testing the recording protocol

In the mouse retina, glycine is released by small-field ACs, which relay inhibition vertically across the inner plexiform layer and thus are involved in cross-over inhibition (**Diamond, 2017**; **Weiss et al., 2008**). Blocking cross-over inhibition between the On- and Off-pathways is therefore expected to have effects on many RGC circuits, which is why we chose the glycine receptor antagonist strychnine to test our recording protocol. Specifically, we focused on strychnine unmasking responses to the other stimulus polarity (**Farajian et al., 2011**). Indeed, we found that strychnine revealed additional On-response components in Off (e.g. G$_1$, G$_2$, G$_4$, G$_6$) and On-Off RGCs (e.g. G$_{11}$, G$_{12}$) RGCs, as can be seen, for instance, in their leading-edge response to the moving bar (**Figure 3—figure supplement 2**). In On RGCs, we did not detect additional (Off) response components. Instead, some On

RGCs exhibited slightly more sustained responses to light increments (e.g. $G_{18}$, $G_{20}$, $G_{22}$). Together, the strychnine-dataset demonstrates that we can resolve drug-related effects on light responses at the RGC type-level.

## Certain RGC types display adaptational response changes

To test if our recording conditions were stable and to exclude major batch effects, we first compared the responses of the control-datasets (Ctrl 1 vs. Ctrl 2). To this end, we computed the difference between the Ctrl 1 and Ctrl 2 mean responses ($\Delta$Ctrl: $\Delta R_{Ctrl2-Ctrl1}$) to chirp and moving bar stimuli for each cell of every RGC type. This allowed us to quantify if and how the responses changed over the time course of an experiment (cf. protocol in *Figure 1e*). Here, we only considered RGC types with >10 sequentially recorded cells (21/32).

Surprisingly, while the majority of RGC types featured stable responses (e.g. $G_1$, $G_{21}$; *Figure 3a*), a substantial number of RGC types (9/21) changed their responses to chirp and/or moving bar stimuli in the absence of any pharmacological perturbation in a highly reproducible manner (*Figure 3—figure supplement 1a and b*). For instance, for Ctrl 2, $G_{23}$ showed reduced responses, whereas $G_{31}$ showed an increased response activity (*Figure 3b*). Interestingly, cells assigned to the functional groups of 'Off' RGC types displayed stable responses, whereas 'On-Off', 'Fast On', 'Slow On', and 'Uncertain RGCs' included types with changing responses (50% (2/4), 34% (1/3), 67% (4/6), and 67% (2/3), respectively). This diversity argues against a systematic effect (such as, e.g., general run-down) and for a cell type-specific phenomenon, which in the following we refer to as 'adaptational response changes'.

## NO affects retinal output in a highly type-specific manner

Next, we investigated the effects of NO on the RGC responses. As with the control-dataset, we computed the cell-wise response differences between Ctrl 1 and NO responses ($\Delta$DETA/NO: $\Delta R_{NO-Ctrl1}$). Similar to the control-dataset, the majority of RGC types displayed stable responses (e.g. $G_2$, $G_{17}$; *Figure 3c*), while ~43% changed their responses significantly (e.g. $G_{28}$, $G_{32}$; *Figure 3d*) following the NO perfusion. We found that the percentage of changing types per functional group was similar to that in the control-dataset: 'Off' (0% (0/5)), 'On-Off' (50% (2/4)), 'Fast On' (34% (1/3)), 'Slow On' (66% (4/6)), and 'Uncertain RGCs' (66% (2/3)). This raised the question if the observed changes in the NO-dataset indeed reflected NO-induced modulations or mostly adaptational response changes (as observed in the control-dataset). We, therefore, tested for each RGC type if the response changes observed for control ($\Delta$Ctrl: $\Delta R_{Ctrl2-Ctrl1}$) and NO ($\Delta$DETA/NO: $\Delta R_{NO-Ctrl1}$) were significantly different (*Figure 3e*). To our surprise, this was only the case for two types: (1) $G_{32}$ ('Off suppressed 2') RGC, which is characterized by a high baseline activity that is strongly suppressed below baseline during light increments and displays increased activity during light decrements, and (2) $G_{18}$, which is considered as being a 'Fast On' type having a response to light increments with a fast response kinetic. This suggests highly type-selective NO effects – at least for temporal responses to chirp and moving bar stimuli.

Next, we leveraged the control-dataset to disentangle NO-induced from adaptational effects at the level of the response features. To this end, we subdivided the chirp stimulus into six feature segments ((1) On, (2) Off, (3) low frequency, (4) high frequency, (5) low contrast, (6) high contrast), and the moving bar into two ((7) On, (8) Off) (*Figure 4c*). Then, for every cell type and every response feature, we computed the difference of the mean responses between the first and second recording separately for the control (*Figure 4a*; left panel) and NO-dataset (*Figure 4a*; middle panel). To isolate NO-induced effects, we computed the differences ($\Delta R_{NO-Ctrl1}$ - $\Delta R_{Ctrl2-Ctrl1}$; *Figure 4a*; right panel), based on the assumption that the adaptational component of the changes would be similar for both datasets. Through this analysis, we found three response behaviors across the RGC types: (1) not NO-affected/stable, (2) showing adaptational cell type-specific changes, and (3) NO-affected (*Figure 4b*). As before, we found that $G_{32}$ is strongly affected by NO; it showed barely any adaptational response changes, yet its activity increased during NO application (*Figure 4b*, (3)). This increase in activity was statistically significant for the chirp's On (1) and Off steps (2) and during both the frequency and contrast modulations ((3)–(6)), as well as for the moving bars trailing edge (8) (for tests, see *Figure 4* legend), suggesting that NO reduces the cell's inhibition by temporal contrast.

Interestingly, a cell type from the same larger functional group that features analogous response properties, $G_{31}$ ('Off suppressed 1'), displayed a similar behavior during the NO application (*Figure 4a*).

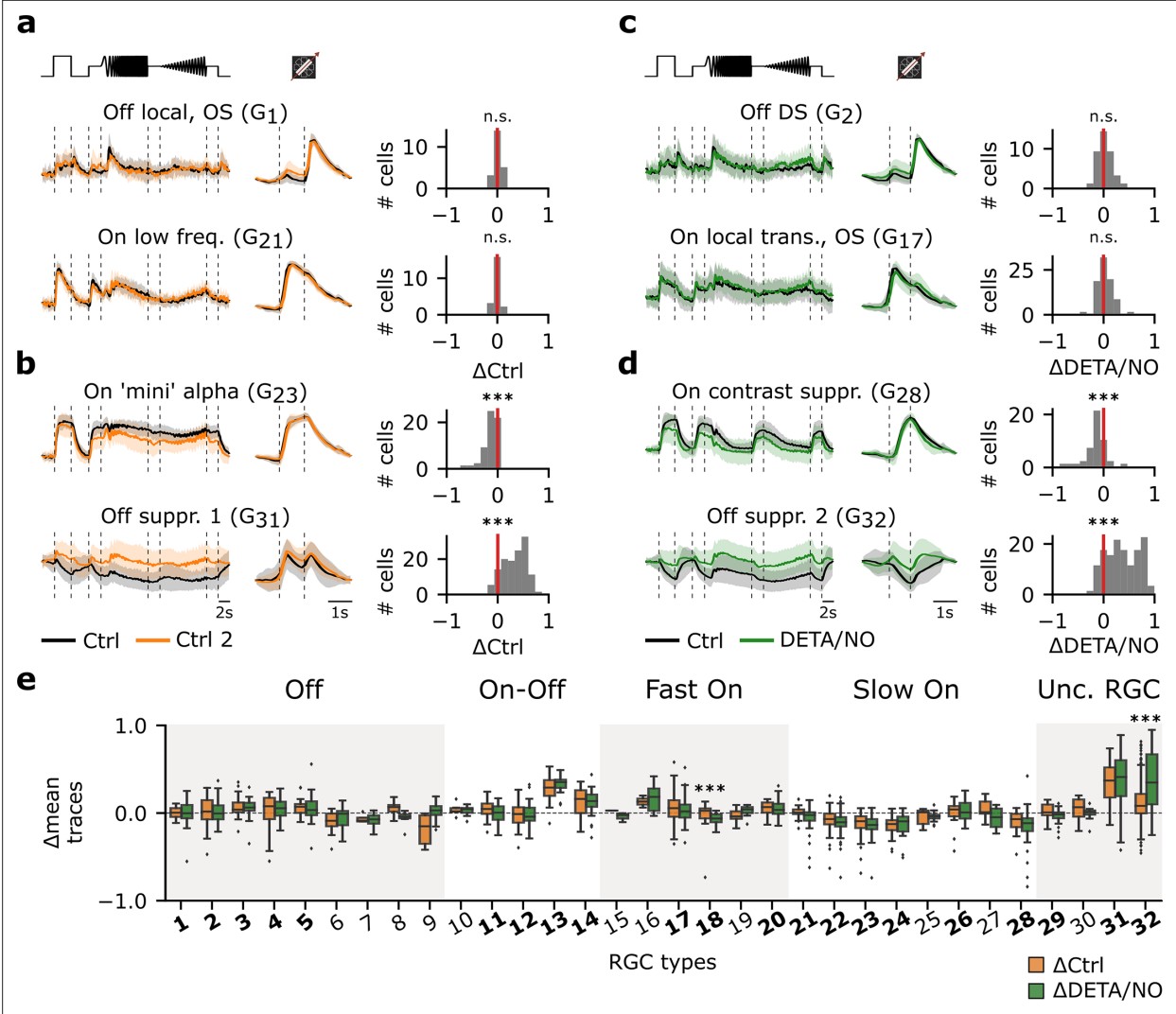

**Figure 3.** Certain retinal ganglion cell (RGC) types are affected by adaptational and/or nitric oxide (NO)-induced effects, while others are unaffected. (**a**) Left: Two representative mean $Ca^{2+}$ responses of sequentially recorded RGC types showing no differences between Ctrl 1 (black) and Ctrl 2 (orange) (top: $G_1$; bottom: $G_{21}$). Right: Corresponding histograms displaying the differences between the average traces of the sequentially recorded cell of the respective cell types. Zero indicates no difference between the response of the same cell across both recordings, whereas negative values indicate a decreased and positive values an increased activity. n.s.: not significant; ***: p<0.001; one-sample t-test. (**b**) Two representative RGC types that show decreased (top: $G_{23}$) and increased (bottom: $G_{31}$) response activity during Ctrl 2. n.s.: not significant; ***: p<0.001; one-sample t-test. (**c**) As in (**a**), but between sequentially recorded Ctrl 1 (black) and DETA/NO (green) (top: $G_2$; bottom: $G_{17}$). n.s.: not significant; ***: p<0.001; one-sample t-test. (**d**) As (**c**), but showing two cell types that display a decreased (top: $G_{28}$) and increased (bottom: $G_{32}$) activity when perfused with DETA/NO. n.s.: not significant; ***: p<0.001; one-sample t-test. (**e**) Box plots of trace differences of all sequentially recorded cells of all RGC types from control- ($\Delta$Ctrl: $\Delta R_{Ctrl2\text{-}Ctrl1}$; orange) and NO-dataset ($\Delta$DETA/NO: $\Delta R_{NO\text{-}Ctrl1}$; green). Bold numbers indicate RGC types with >10 sequentially recorded cells per dataset and condition. Dashed line shows zero baselines, i.e., no difference between traces. Diamond symbols represent outliers. Gray and white background blocks summarize the larger functional groups for better visualization (Off, On-Off, Fast On, Slow On, Uncertain RGCs). ***: p<0.001; Mann-Whitney U-test.

The online version of this article includes the following figure supplement(s) for figure 3:

**Figure supplement 1.** Adaptational, cell type-specific response changes without pharmacological perturbation.

**Figure supplement 2.** Testing the effects of strychnine on different retinal ganglion cell (RGC) type responses.

However, $G_{31}$ showed this response change for most features already in the control, suggesting that this effect was primarily adaptational (*Figure 4a*, right panel). Still, our analysis suggests that NO had a significant effect, though different from what we observed for $G_{32}$: in $G_{31}$ the response to the chirp's Off step (2) increased, whereas that to the moving bar's leading edge (7) decreased. Note that

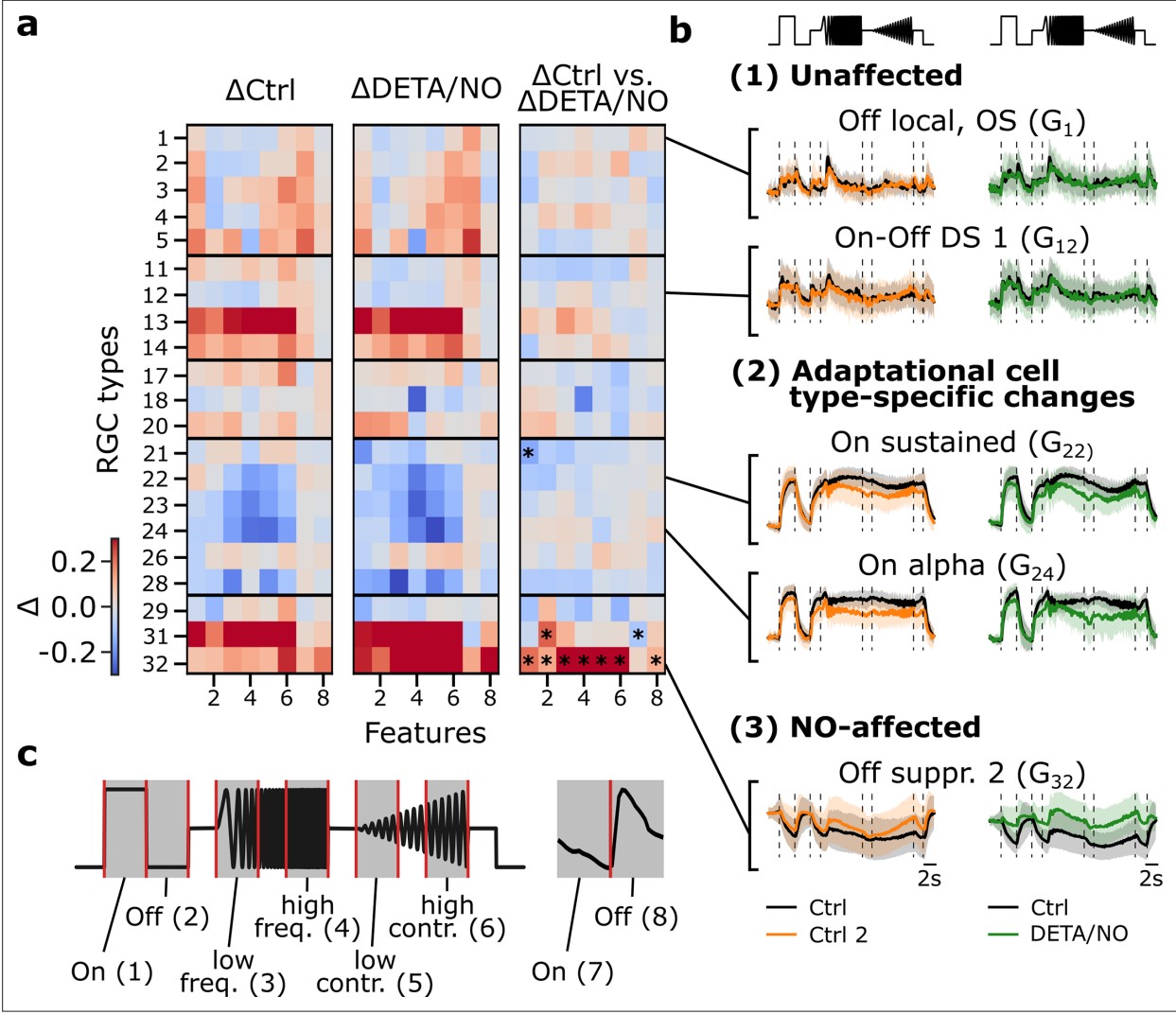

**Figure 4.** Disentangling nitric oxide (NO)-induced effects from adaptational response changes reveals type-specific NO modulation. (**a**) Left: Difference between sequentially recorded Ctrl 2 and Ctrl 1 retinal ganglion cell (RGC) traces per type subdivided into eight features ($\Delta$Ctrl: $\Delta R_{Ctrl2-Ctrl1}$). Color code indicates response increase (red), no change (white), and decrease (blue) for Ctrl 2. Middle: Difference between DETA/NO and Ctrl ($\Delta$DETA/NO: $\Delta R_{NO-Ctrl1}$). Right: Difference between the two heat maps ($\Delta R_{NO-Ctrl1} - \Delta R_{Ctrl2-Ctrl1}$). Asterisks indicate significant differences of the trace differences of all cells per feature and per cell type between $\Delta$Ctrl and $\Delta$DETA/NO using independent two-sided t-test and Bonferroni correction for multiple tests; *: $p < 0.0003$. (**b**) Example chirp traces are categorized into unaffected (top two types: $G_1$, $G_{12}$), adaptational (two middle types: $G_{22}$, $G_{24}$), and NO-affected (bottom: $G_{32}$). Left traces show exemplary responses per type from the control-dataset (black: Ctrl; orange: Ctrl 2) and NO-dataset (black: Ctrl; green: DETA/NO). (**c**) Subdividing the chirp (left) and moving bar (right) stimuli into eight features for detailed feature analysis. The chirp is subdivided into six features ((1) on, (2) off, (3) low frequency, (4) high frequency, (5) low contrast, and (6) high contrast); the moving bar into two ((7) on and (8) off).

it is possible that additional NO effects on $G_{31}$ may have been 'masked' by the strong adaptational response changes.

In the previous analysis, $G_{18}$ showed significant differences between the control- and NO-dataset, but this effect was not mirrored by the feature-wise analysis, indicating that a potential effect may be much weaker compared to the effect NO has on $G_{32}$. Notably, RGC types that were assigned to the group of the so-called 'Slow On' types ($G_{21}$-$G_{28}$), which exhibit strong and sustained responses during the frequency and contrast sequences of the chirp stimulus, showed a decrease in activity in both datasets (e.g. $G_{24}$; *Figure 4b*, (2), bottom). Consequently, the changes in these response features ((3)–(6)) are likely adaptational (*Figure 4a*) – as the changes can be found in the control- (*Figure 4a*, left panel) as well as in the NO-dataset (*Figure 4a*, middle panel), but not in the difference (*Figure 4a*, right panel). Our conclusion that both adaptational and NO-mediated effects on RGC responses are

highly cell type-selective is further supported by the fact that we also found several RGC types that showed stable responses during control and NO application (*Figure 4b*, (1)).

Taken together, our analysis revealed that the adaptational and NO-induced effects occurred on a feature-specific as well as cell type-specific level. On the one hand, several 'Slow On' types showed adaptational effects in response to temporal frequency and contrast features, while other features were not affected. On the other hand, at least one distinct RGC type ($G_{32}$) displayed a significantly increased response modulation of most features during the NO application; hence hinting toward an effect of NO on response suppression, as in the case of $G_{32}$ is elicited by temporally changing stimulus contrast. Consequently, we focused on a more in-depth analysis of NO-induced effects on $G_{32}$.

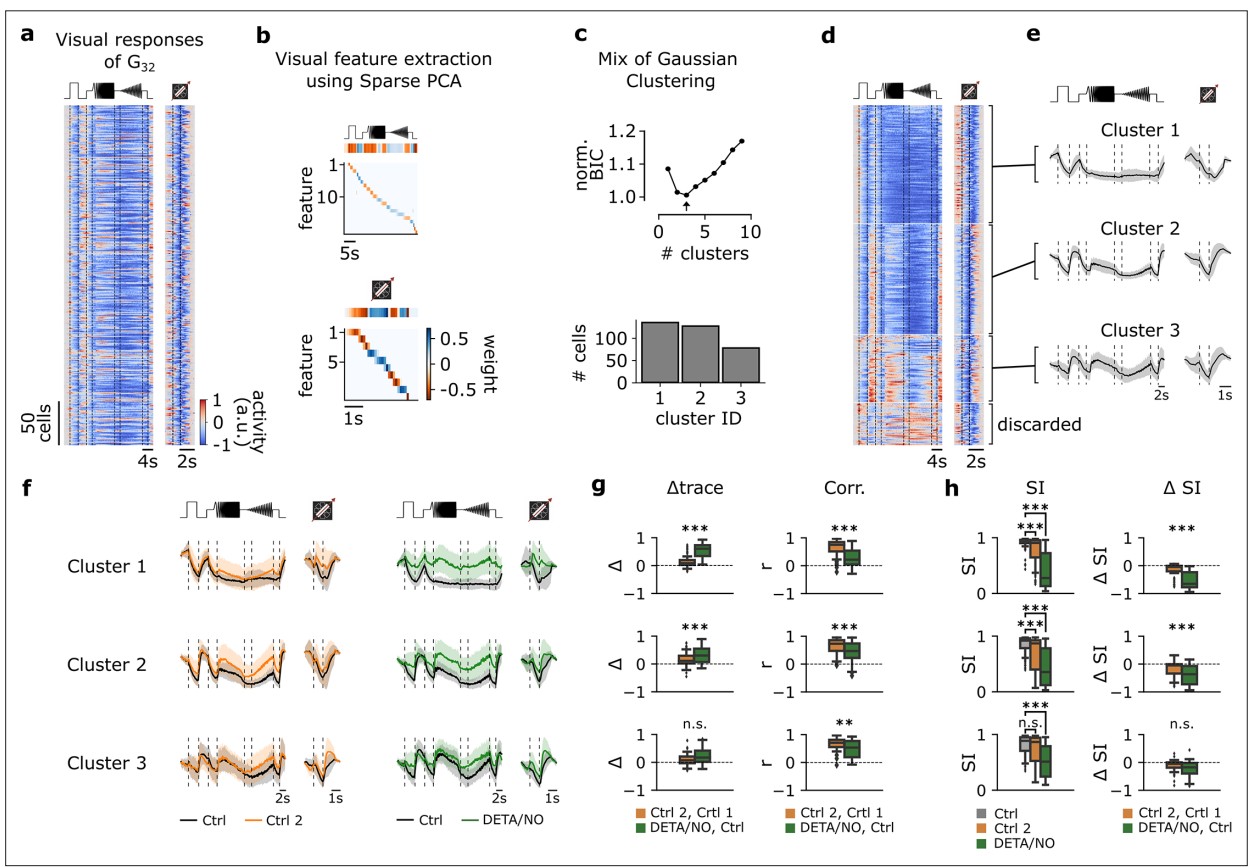

**Figure 5.** Functional clustering of the $G_{32}$ reveals three distinct types that are differently affected by nitric oxide (NO). (**a**) Visual responses of $G_{32}$ cells recorded from several experiments in response to the full-field chirp (left) and moving bar (right) stimuli. (**b**) Visual features extracted from chirp (top) and moving bar (bottom) stimuli using sparse principal component analysis (PCA) on the responses. Color indicates the weight of each feature. (**c**) Top: Bayesian information criterion (BIC) as function of number of clusters. Arrow indicates the lowest BIC and the number of clusters to choose. Bottom: Number of cells per predicted cluster. (**d**) Cells sorted according to their assigned cluster. Cells at the bottom were discarded. (**e**) Mean responses of the three corresponding clusters for the chirp (left) and moving bar (right). (**f**) Left: Sequentially recorded mean responses of the three clusters to Ctrl 1 (black) and Ctrl 2 (orange). Right: Cluster mean responses to Ctrl 1 (black) and DETA/NO (green). (**g**) Left: Trace difference between Ctrl 2 and Ctrl 1 (orange) and DETA/NO and Ctrl (green) for the three clusters (clusters 1–3 from top to bottom). Right: Correlation coefficient between Ctrl 2 and Ctrl 1 (orange) and DETA/NO and Ctrl 1 (green) for the three clusters. n.s.: not significant; **: p<0.01, ***: p<0.001; independent t-test and Mann-Whitney U-test. (**h**) Left: Suppression index (SI) computed for Ctrl 1 (gray), Ctrl 2 (orange), and DETA/NO (green) for the three clusters. n.s.: not significant; **: p<0.01, ***: p<0.001; Kruskal-Wallis test and Dunnett's test. Right: Difference of SI between Ctrl 2 and Ctrl 1 (orange) and DETA/NO and Ctrl 1 (green). n.s.: not significant; **: p<0.01, ***: p<0.001; independent t-test and Mann-Whitney U-test.

The online version of this article includes the following figure supplement(s) for figure 5:

**Figure supplement 1.** Further evaluation of the $G_{32}$ functional clustering.

# Clustering of $G_{32}$ responses reveals three functionally distinct RGC types with different NO-sensitivity

According to *Baden et al., 2016*, $G_{32}$ features a coverage factor of ~4. As the average coverage factor of mouse RGCs was estimated to be ~2 (*Baden et al., 2016*; *Bae et al., 2018*), $G_{32}$ likely consists of several (functional) RGC types. This is in line with the high variation of $G_{32}$ responses, which also supports the presence of multiple functional types (see *Figure 3e*).

To test this, we performed Mixture of Gaussians clustering of the RGCs assigned to $G_{32}$ (*Figure 5a–c*) using the Ctrl 1 responses to chirp and moving bar stimuli from both datasets (*Figure 5a*). Since the normalized Bayesian information criterion (BIC; *Figure 5c*, top; see Methods) values were close for n=2 and n=3 clusters, we used further tests to determine the optimal cluster number (*Figure 5—figure supplement 1a and b*). These showed that for n=3 the intra-cluster correlations were higher and more consistent across clusters than for n=2 (*Figure 5—figure supplement 1c*). Therefore, we concluded that $G_{32}$ likely contains three distinct response types – all suppressed-by-contrast (SbC) but to different degrees (*Figure 5d and e*). All three $G_{32}$ clusters showed little adaptation for the control-dataset, but displayed differential modulations in response to NO application, with cluster 1 exhibiting the strongest NO effect (*Figure 5f and g*, left) for both stimuli. The NO effect was statistically significant in clusters 1 (n=134) and 2 (n=126) – both for mean trace difference and correlation (*Figure 5g*, left and right, respectively) – but only for trace correlation in cluster 3 (n=77).

Since the prominent feature of these RGC types is suppression by temporal stimulus contrast, we compared their suppression strength between the conditions using a suppression index (SI; see Methods). Here, we found that in cluster 1, SI marginally, yet significantly, changed between Ctrl 1 and Ctrl 2, but was strongly and significantly reduced by NO as ΔSI were significantly different (*Figure 5h*, top panel). Similar response behavior can be found in cluster 2, which displayed significant differences in SI and ΔSI in both control- and NO-dataset (*Figure 5h*, middle panel). In fact, with NO, these cells lost their suppressive feature and were rather excited than suppressed by the stimuli. In contrast,

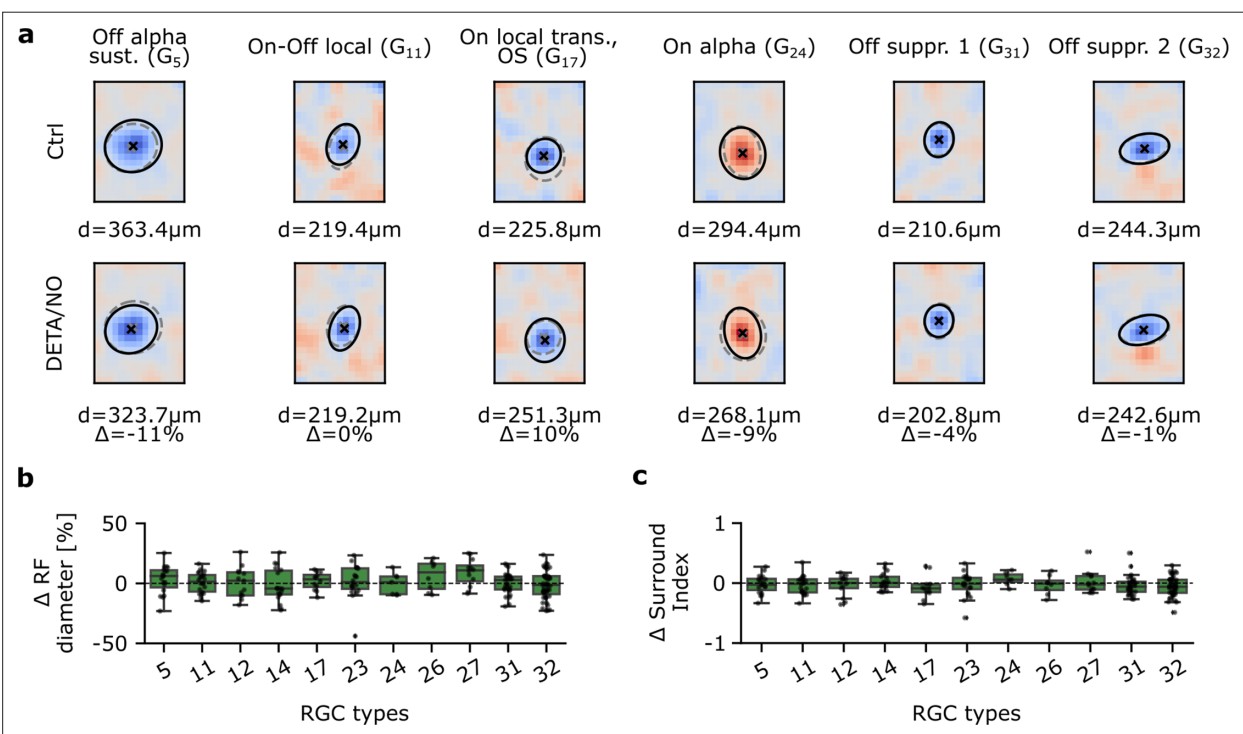

**Figure 6.** Spatial receptive fields (sRFs) are not affected by nitric oxide (NO) across various retinal ganglion cell (RGC) types. (**a**) Representative estimated sRFs of six RGC types. Top: Estimated sRFs to Ctrl 1. Cross indicates RF center; solid line indicates outline of the Gaussian fit of the RF center; dashed outline indicates corresponding Gaussian fit of the same cell to DETA/NO. Bottom: Same as top, but of DETA/NO condition. (**b**) RF diameter difference in percentage between DETA/NO and Ctrl 1. Only types with more than five sequentially recorded cells were included. One-sample t-test. (**c**) Surround index difference between DETA/NO and Ctrl 1. Only types with more than five sequentially recorded cells were included. One-sample t-test.

cluster 3 showed no significant differences of the SI between Ctrl 1 and Ctrl 2, but a clear modulation by NO (*Figure 5h*, bottom panel).

Taken together, our data indicate that $G_{32}$ may consist of three SbC RGC types and that in at least two of them, the contrast suppression is strongly modulated by NO.

## NO does not affect RGC RF properties

So far, we focused on effects in the temporal response domain, where we found that mainly $G_{32}$ types were affected by an increase in NO levels. However, NO has been shown to affect electrical coupling, e.g., by reducing conductance between AII ACs and On-cone BCs (*Mills and Massey, 1995*), and hence may alter RF properties. Therefore, we next investigated the effects of NO on the sRFs of the individual RGC types. To this end, following the same experimental paradigm as described earlier, we recorded RGC responses to binary dense noise. Next, we computed their sRFs for both recording conditions (*Figure 6a*) using spike-triggered averaging (*Chichilnisky, 2001*), obtaining control and NO sRFs, and then fitted a Gaussian to each sRF's center (*Figure 6a*). We focused the following analysis on RGC types with reliable sRF estimates (see Methods). These included types in all five larger RGC groups (*Baden et al., 2016*): $G_5$ for 'Off'; $G_{11}$, $G_{12}$, and $G_{14}$ for 'On-Off'; $G_{17}$ for 'Fast On'; $G_{23}$, $G_{24}$, $G_{26}$, and $G_{27}$ for 'Slow On'; $G_{31}$ and $G_{32}$ for 'Uncertain RGCs'. In these types, sRFs were very stable in both control and NO condition.

Using the difference in sRF center diameter between control and NO as a metric (*Figure 6b*), we did not find NO to cause any significant changes in sRF size in any of the analyzed RGC types. Next, we tested if the sRF surround was affected by NO, because a modulation of inhibitory synaptic input and/or electrical coupling may cause a change in surround strength. As a measure of surround strength, we computed a surround index (see Methods) of control and NO sRFs (*Figure 6c*). Like with the sRF center diameter, we did not find significant differences for any of the analyzed RGC types. Surprisingly, also $G_{32}$ did not show NO-mediated differences in its sRF properties, implying that NO may only affect its temporal response, but not its RF organization.

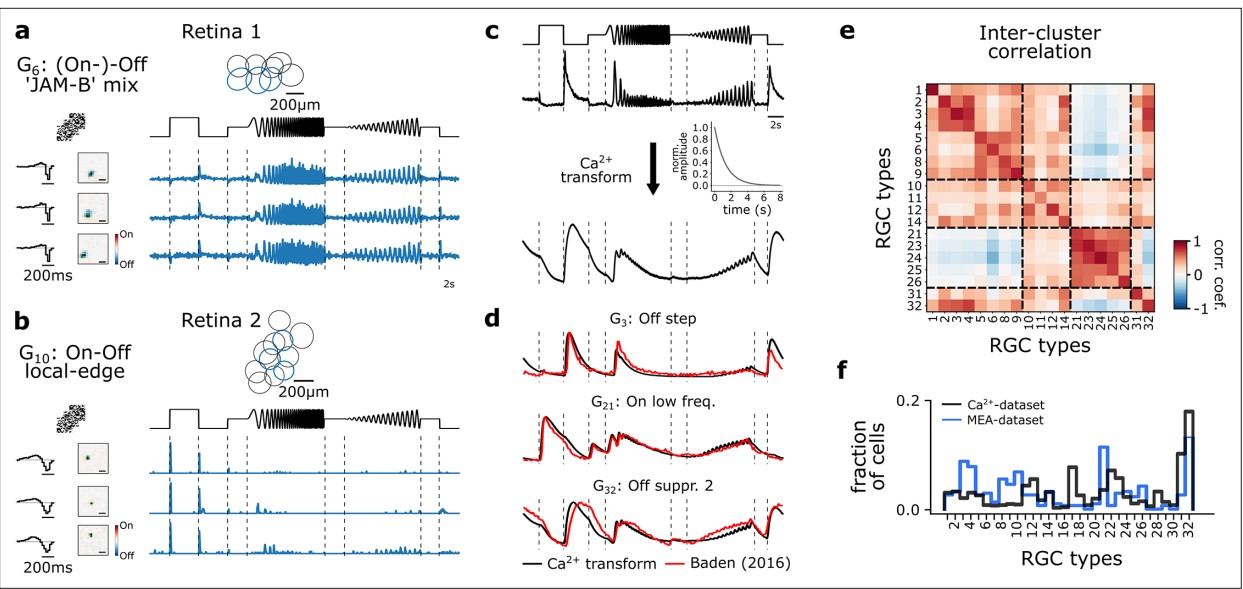

**Figure 7.** Multi-electrode recordings and pseudo-calcium transformation. (**a**) Representative retinal ganglion cell (RGC) type, $G_6$ '(On-)Off 'JAM-B' mix, recorded from one retina forming a mosaic. Three exemplary response traces of the same type. Cells are indicated in the mosaic as blue. Left: tSTAs and sSTAs computed from their responses to a checkerboard stimulus. Right: peristimulus time histograms (PSTHs) to the chirp stimulus. (**b**) As (**a**), but displaying another type, $G_{10}$ 'On-Off local edge'. (**c**) Illustration of the transformation of PSTHs to pseudo-calcium traces using an OGB-1 filter described in *Baden et al., 2016*, to match and identify RGC types. (**d**) Overlay of representative mean responses of three RGC types of the *Baden et al., 2016*, dataset (red) and the assigned $Ca^{2+}$-transformed (black) traces to the chirp stimulus. (**e**) Inter-cluster correlation matrix of PSTHs of each assigned RGC type within a group with its group mean. Only types with n>5 cells per type were included. Dashed boxes indicate functional groups (Off, On-Off, Slow On, and Uncertain RGCs; see *Baden et al., 2016*). Color bar indicates Pearson correlation coefficient. (**f**) Comparison of the distribution of predicted RGC types between the MEA- (blue) and $Ca^{2+}$-datasets (black).

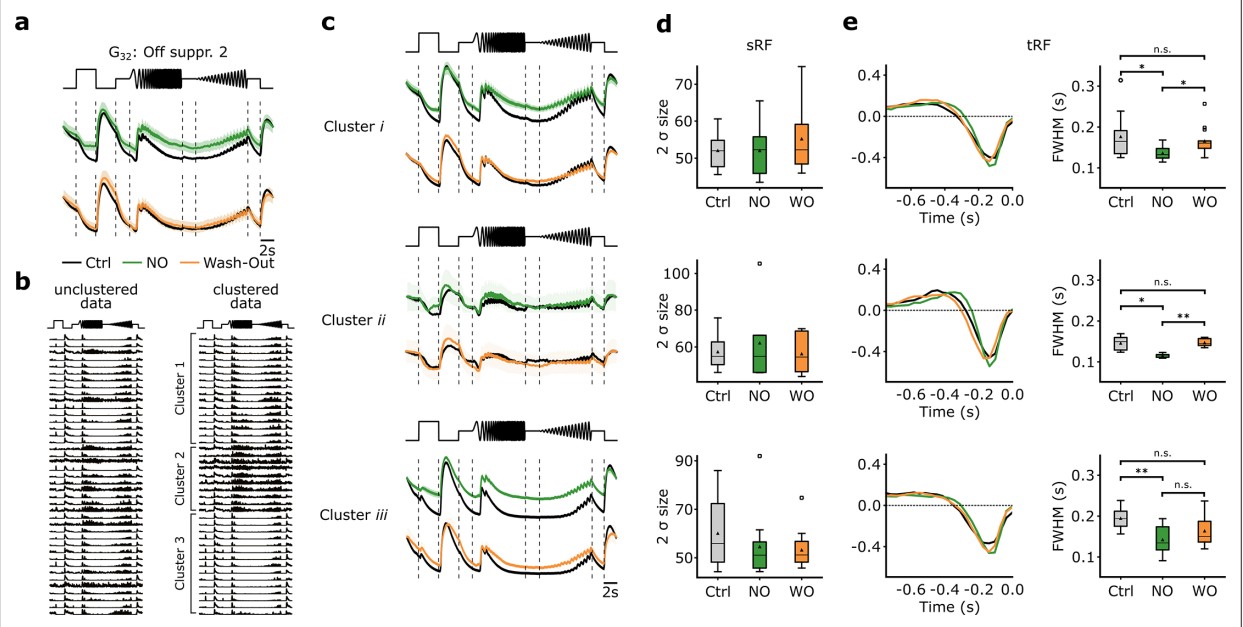

**Figure 8.** Nitric oxide (NO) only affects temporal features of three distinct clusters of $G_{32}$. (**a**) Mean $Ca^{2+}$-transformed responses to the chirp of the retinal ganglion cell (RGC) type $G_{32}$. Top: Sequentially recorded RGC responses to the Ctrl (black) and DETA/NO (green) conditions. Bottom: Sequentially recorded RGC responses to the Ctrl (black) and Wash-Out (orange) conditions after DETA/NO application. (**b**) Left: Unclustered peristimulus time histograms (PSTHs) of cells assigned to type $G_{32}$. Right: Cell's PSTHs were clustered and sorted into three distinct clusters. (**c**) Sequentially recorded mean responses of three clusters to Ctrl (black), DETA/NO (green), and Wash-Out (orange). (**d**) Ellipse size of the fitted Gaussian of the spatial receptive field (sRF) of the three conditions (Ctrl: black; DETA/NO: green; Wash-Out: orange) for the three clusters (clusters *i–iii*; top to bottom). All tested conditions were not significant; two-sided t-test. (**e**) Left: temporal receptive field (tRF) kernels of the three conditions (Ctrl: black; DETA/NO: green; Wash-Out: orange) for three clusters (clusters *i–iii*; top to bottom). Right: Full width at half minimum (FWHM) of the temporal RF kernels of the three conditions (Ctrl: black; DETA/NO: green; Wash-Out: orange) for the three clusters. *: p<0.05; **: p<0.01, repeated measures ANOVA and Dunnett's test.

The online version of this article includes the following figure supplement(s) for figure 8:

**Figure supplement 1.** Bayesian information criterion (BIC) as a function of number of clusters for $G_{32}$ in the multi-electrode array (MEA)-dataset.

**Figure supplement 2.** Pseudo-calcium and peristimulus time histograms of $G_{32}$ clusters.

**Figure supplement 3.** Correlating $G_{32}$ clusters of the $Ca^{2+}$-dataset and multi-electrode array (MEA)-dataset.

Taken together, at least for the tested RGC types, we did not detect any significant NO effects neither on sRF center size nor surround strength.

## NO affects the temporal response kinetics of $G_{32}$ subtypes

The high spatial resolution of two-photon imaging allowed us to record and identify individual RGCs. However, it is limited by having a low temporal resolution to capture subtle temporal response dynamics, which led us to record a complementary dataset using MEA recordings. Since we found that the temporal response domain of $G_{32}$ and its subtypes were affected by NO, MEA recordings enabled us to further investigate subtle neuromodulatory effects with a higher temporal resolution.

In particular, we recorded light-induced RGC responses (n=391) from four retinae under three consecutive conditions similar to the $Ca^{2+}$-dataset: (1) control, (2) NO-donor (DETA/NO; 100 μM), and (3) wash-out. As for the $Ca^{2+}$-dataset, we used the chirp stimulus for type identification as well as a checkerboard stimulus to estimate RFs. To identify and compare RGC types across the $Ca^{2+}$- and MEA-dataset, we clustered the RGC responses across retinae (*Figure 7a and b*) and transformed their spike trains in response to the chirp stimulus into pseudo-calcium traces as described in *Goldin et al., 2022* (*Figure 7c*; see Methods). Next, we matched the pseudo-calcium traces with the RGC types obtained from *Baden et al., 2016* (*Figure 7d and e*). Overall, we were able to record responses from all known RGC types, yet the sampling fractions for most types differ between the datasets, which can be explained by the recording bias inherent in MEA (*Trapani et al., 2023*; *Figure 7f*).

We focused the following analysis on RGCs classified as $G_{32}$ (n=42). We found that, similar to the $Ca^{2+}$-dataset (*Figure 5*), $G_{32}$ showed reduced suppression during NO application (*Figure 8a*, top), which was not present anymore in the recording after wash-out, indicating that the NO-induced effect may be reversible (*Figure 8a*, bottom). Since we found in our $Ca^{2+}$-dataset that $G_{32}$ consists of three subtypes, we applied the same clustering approach to the electrically recorded RGCs assigned to $G_{32}$, revealing also three clusters (*Figure 8b*, *Figure 8—figure supplement 1*). For all three clusters, we found NO-induced response modulations but to varying degrees. The NO effects were at least partially reversible (see wash-out condition; *Figure 8c*).

We also performed an analysis directly at the spike rate level from the peristimulus time histograms (PSTH) (*Figure 8—figure supplement 2*). We computed the cumulative firing rate for four time windows (features) of the chirp response, where the $Ca^{2+}$-dataset suggests suppression (*Figure 8—figure supplement 2a and b*). We found for all three clusters an increase in cumulative firing rate during DETA/NO application – indicative of a reduction in suppression – for at least one feature (*Figure 8—figure supplement 2c*). That the NO effect was differential (e.g. compare 'Freq. step' and 'On step' between clusters) further supports the presence of multiple RGC types in $G_{32}$.

Based on the $Ca^{2+}$-dataset, we did not detect significant effects of NO on RFs. However, as recording time, and hence stimulus presentation time, is limited per scan field, we repeated the RF analysis for the MEA-dataset. MEA recordings allow for much longer dense noise stimuli and, thus, can yield more precise RF estimates. In line with the $Ca^{2+}$-dataset, we did not find any NO-induced modulation of sRF size in any of the clusters (*Figure 8d*). However, the temporal RF (tRF) kernels showed significantly faster response kinetics during NO application for all three clusters (*Figure 8e*). Also, the effects were reversible for two clusters (except for cluster *iii*). That we did not detect this effect in the $Ca^{2+}$-dataset is likely due to the lower temporal resolution of the $Ca^{2+}$ recordings. Note that the $G_{32}$ subtypes identified in both datasets do not necessarily correspond to the same RGC types: While two clusters show high correlations between $Ca^{2+}$ and MEA-data (clusters 2 vs. *ii*, 3 vs. *i*), one clearly differs in its temporal response (cluster 1 vs. *iii*; *Figure 8—figure supplement 3a and b*; see also Methods).

Taken together, our MEA-dataset confirms that $G_{32}$ consists of three subtypes and that their temporal responses are modulated by NO: it reduced their suppression by temporal contrast. Additionally, the analysis of the MEA-dataset (tRF) revealed that all three subtypes displayed faster response kinetics under NO, an effect that was reversible. The sRFs were unchanged by NO, in line with the $Ca^{2+}$ data.

## Discussion

We used two-photon $Ca^{2+}$ imaging and MEA recordings to measure RGC responses to various visual stimuli to investigate the neuromodulatory effects of elevated NO levels on signal processing across RGC types in the mouse retina. To our surprise, even without pharmacological perturbation, we found that about one-third of the RGC types displayed highly reproducible and cell type-specific response changes during the course of an experiment – a finding of potentially high relevance especially for pharmacological experiments in the ex vivo retina using two-photon $Ca^{2+}$ imaging. Accounting for these adaptational changes enabled us to isolate NO-related effects on RGC responses. Here, we revealed that mainly the RGCs assigned to $G_{32}$ ('Off suppressed 2') were affected by NO, which strongly reduced the response suppression and rendered the cells more active. Further, we demonstrated that $G_{32}$ likely consists of three types – consistent with its high coverage factor (*Baden et al., 2016*) – that were all differentially modulated by NO. Finally, for a representative subset of RGC types, we showed that elevating the NO level had no discernible effect on sRF size or surround strength. In addition, we were able to confirm those results using MEA recordings and showed that NO caused faster temporal response kinetics of these three subtypes and partially increased firing rates; these changes were mostly reversible. Together, our data suggest that NO specifically modulates response suppression and kinetics in a group of contrast-suppressed RGC types. Additionally, our study demonstrates the need for recording paradigms that take adaptational, non-drug-related response changes into account when analyzing potentially subtle pharmacological effects.

## NO as a (neuro-)modulator in the retina

nNOS has been detected in different retinal cell classes (*Blom et al., 2009*; *Blute et al., 2000*; *Blute et al., 1997*; *Bredt et al., 1990*; *Dawson et al., 1991*; *Eldred and Blute, 2005*; *Palamalai et al., 2006*; *Yamamoto et al., 1993a*; *Yamamoto et al., 1993b*) and the main NO-sensor (NO-GC), which connects NO to intracellular cGMP signaling (*Blom et al., 2012*; *Blute et al., 1998*; *Eldred and Blute, 2005*; *Gotzes et al., 1998*), is present in all retinal layers (*Vielma et al., 2014*). This and earlier findings of NO modulating the response gain of BCs (*Snellman and Nawy, 2004*; *Tooker et al., 2013*) and RGCs (*Wang et al., 2003*) suggest that NO constitutes a neuromodulatory system within the retina involved in light adaptation. However, in the light-adapted mouse retina, light stimulus-dependent NO production seems to mainly occur in specific AC types, with the nNOS-2 AC being the main source of endogenous NO (*Jacoby et al., 2018*). In particular, it releases NO in response to flickering light (i.e. fast changes in contrast) (*Blom et al., 2012*; *Eldred and Blute, 2005*; *Jacoby et al., 2018*; *Vielma et al., 2012*; *Wang et al., 2007*). Thus, it has been proposed that nNOS-2 ACs report fast contrast changes at photopic conditions (*Jacoby et al., 2018*; *Vielma et al., 2012*). This points at an interesting though speculative functional role of the NO-sensitivity in $G_{32}$, namely that NO helps 'highlighting' *changes* in contrast by reducing suppression and accelerating response kinetics. Thereby, $G_{32}$ RGCs could relay contrast changes to higher visual targets.

Interestingly, this RGC type-selectivity of NO is reminiscent of a recent study, where the neuromodulator DA was found to modulate distinct response features of specific RGC subtypes (*Warwick et al., 2023*). What cellular mechanisms underlie the action of NO on $G_{32}$ RGCs, or its upstream circuit, will be interesting to investigate in future studies.

To ensure reliable RGC type classification, our set of visual stimuli was restricted to artificial ones (full-field chirps, moving bars, and dense noise). Such artificial stimuli probe the stimulus space in a rather selective and limited fashion. Hence, we cannot exclude that we missed NO effects that may have become apparent for other, more complex stimuli. Specifically, natural images or movies – stimuli that are closer to what the retina evolved to process (*Qiu et al., 2021*) – may be needed for a more complete picture of the functional implication of retinal neuromodulation.

That natural stimuli can reveal novel nonlinear properties of retinal functions were demonstrated, for example, by *Goldin et al., 2022*, who showed that an RGC's contrast selectivity can be context-dependent. Similarly, *Höfling et al., 2024*, recorded natural movie-evoked RGC responses to train a convolutional neural network, which through in silico experiments allowed them to discover that transient SbC RGC ($G_{28}$) feature center color-opponent responses and may signal context changes. These studies highlight that future studies on retinal neuromodulators should also employ natural stimuli.

Another important aspect to consider when studying NO neuromodulation in the retina is the level of light adaptation. Several studies proposed that NO facilitates the transition across light levels (*Mills and Massey, 1995*; *Snellman and Nawy, 2004*; *Tooker et al., 2013*), especially to photopic conditions (*Jacoby et al., 2018*). Since we employed two-photon imaging, which inevitably results in a certain level of background 'illumination' (see discussion in *Euler et al., 2009*), our experiments were performed in the low photopic range. Therefore, NO-mediated neuromodulation may serve additional light-level dependent functions: More globally during the transition from scotopic to mesopic/photopic, and more cell type-specific in the photopic regime, as we reported here.

Finally, a gaseous neuromodulator like NO poses a additional problem when studying its effects: A donor is needed and the final concentration of the neuromodulator delivered to the cells depends on many factors (*Beckman and Koppenol, 1996*). We used the NO-donor DETA/NO, because its long half-life time ($t_{1/2}$) of > 20 hr enables a steady delivery of NO within a tissue. Assuming for NO in tissue a $t_{1/2}$ of 2 min, a freshly prepared DETA/NO solution of 100 μM is expected to release about 0.25 μM NO (*Ramamurthi and Lewis, 1997*). This is a conservative estimate; for longer NO half-life times, the concentrations would be higher. Estimates for the endogenous NO concentration in the retina range from a few 100 nM at the RGCs (*Kalamkarov et al., 2016*) to ~1 μM at the vitreous boundary (*Guthrie, 2014*; *Guthrie and Mieler, 2014*). Therefore, the here applied concentration results in a small to moderate NO increase within the measured physiological range. The fact that this already resulted in a clear, type-specific effect, argues for NO having a potent neuromodulatory effect on the retinal output.

It has been reported for DETA/NO that the donor itself – independent of NO – may affect the electrical properties of cultured cerebellar granule cells by reversibly activating a cation-selective

channels (*Thompson et al., 2009*). While we cannot exclude that this side-effect of DETA/NO may have contributed to the effects we observed in $G_{32}$ RGCs, we consider this unlikely, because substantial side-effects were mainly observed by *Thompson et al., 2009*, at much higher DETA/NO concentrations (3 mM) than we used in our study.

## SbC cells in the mouse retina

Functionally, RGC types referred to as 'SbC' are characterized by a decrease in their activity for both positive and negative temporal contrasts within their RF (*Jacoby and Schwartz, 2018*; *Jacoby et al., 2015*; *Tien et al., 2016*; *Tien et al., 2015*). In *Baden et al., 2016*, three functional RGC types were labeled SbC based on their light stimulus-evoked $Ca^{2+}$ response being suppressed primarily by positive (On-SbC: $G_{28}$) or negative temporal contrast (Off-SbCs: $G_{31}$, $G_{32}$).

Notably, $G_{32}$ ('Off suppressed 2') is also suppressed by the moving bar stimulus, suggesting that the cells are also sensitive to spatial contrast (i.e. an edge appearing in their RF). Coverage analysis indicated that $G_{32}$ may contain several RGC types (*Baden et al., 2016*) – in line with our cluster analysis. It is still unclear if $G_{32}$ contains one (or more) of the individually studied SbC types in mouse (*Jacoby and Schwartz, 2018*; *Tien et al., 2016*; *Tien et al., 2015*; *Wienbar and Schwartz, 2022*), yet, recently, *Goetz et al., 2022*, speculated if the novel bursty-SbC RGC (*Wienbar and Schwartz, 2022*) aligns with (a sub-cluster of) $G_{32}$.

## Adaptational, cell type-specific response changes

Every recording method introduces technique-specific biases that have to be considered in the data analysis and interpretation. For two-photon imaging with fluorescent $Ca^{2+}$ sensors, these potential biases include $Ca^{2+}$ buffering, sensor bleaching by the excitation laser, and, in the case of bulk loading with synthetic dyes (as in our experiments; see also *Briggman and Euler, 2011*), slow leakage of the indicator from the cells. In retinal imaging, additional potentially confounding factors are an excitation laser-induced baseline activity and photoreceptor bleaching (*Euler et al., 2019*; *Euler et al., 2009*). These biases are expected to be systematic, e.g., causing a general decrease in signal-to-noise ratio across (RGC) responses. To account for this, our recording paradigm produced a control- and a NO-dataset consisting of sequentially recorded RGC responses.

When analyzing the control-dataset, we were surprised by finding response changes in the second control measurement 10 min after the first control measurement in approximately one-third of the RGC types in the absence of the NO-donor. These changes were consistent for a particular type but differed between types. Notably, we did not observe a simple overall decrease or increase in activity, but rather selective changes of response features: For instance, in $G_{24}$ ('Slow On') only the response to high frequency and low contrast was reduced, while the remaining response features were not affected. As mentioned above, the time differences between Ctrl 1 and Ctrl 2 was 10 min and Ctrl and DETA/NO was approximately 25 min. However, we do not think that this has a substantial effect on our results because when a change for either Ctrl 2 or DETA/NO was observed, it followed the same trend – other than in $G_{32}$ (and $G_{18}$).

Together, this strongly argues against a systematic, recording technique-related bias but rather for an adaptational effect. Currently, we can only speculate about the mechanism(s) underlying this type-specific adaptation. The most parsimonious explanation may be related to the ex vivo condition of the retina: While we allowed the tissue to settle and adapt to perfusion medium, light level, temperature, etc. for ~25 min, extracellular signaling molecules, such as neuromodulators, may be depleted and washed out throughout the experiment, resulting in differential adaptation of various RGC types. In any case, as type-selective adaptations can confound the recorded responses in a complex manner, a sequential-recording paradigm as the one described here is recommended – in particular for pharmacological experiments.

## Combining large-scale population recordings, RGC classification, and sequential recordings to study neuromodulation of retinal output

In this study, we investigated the neuromodulatory effects of NO on the retinal output signal. To this end, we combined experimental and computational approaches to dissect NO-mediated effects at the RGC type-level. The latter is important for understanding neuromodulator function for early vision because the visual information is sent to the brain via parallel feature channels, represented by >40

RGC types in the mouse retina (*Baden et al., 2016*; *Bae et al., 2018*; *Goetz et al., 2022*; *Tran et al., 2019*). We demonstrated that our approach enabled us to distinguish adaptational from actual NO-induced effects using a rather simple and straightforward linear analysis (i.e. focusing on mean trace or RF size differences), assuming that the adaptational and NO effects are independent and sum linearly, which means that we may have missed potential nonlinear effects.

The combination of two-photon Ca$^{2+}$ imaging and MEA recordings allowed us to confirm our findings and make use of two methods that complement one another. Finally, the presented pipeline for analyzing neuromodulation in neural circuits constitutes a framework that can be easily extended by applying more advanced analyses.

## Methods

### Animals and tissue preparation for two-photon Ca$^{2+}$ imaging

All animals for the two-photon Ca$^{2+}$ imaging experiments were conducted at the University of Tübingen and were performed according to the laws governing animal experimentation issued by the German Government as well as approved by the institutional animal welfare committee of the University of Tübingen. For all experiments, we used retinae (n=26) from C57Bl/6J mice (n=14; JAX 000664) of either sex between the age of 4 and 16 weeks. All animals were kept in the local animal facility and housed under the standard 12 hr/12 hr day/night cycle at 22°C and a humidity of 55%.

The following procedures were carried out under very dim red (>650 nm) light. Before each imaging experiment, the animal was dark-adapted for >1 hr, then anesthetized with isoflurane (CP-Pharma) and sacrificed by cervical dislocation. Immediately after, the eyes were enucleated with a dorsal cut as orientation landmark and hemisected in carboxygenated (95% O$_2$, 5% CO$_2$) artificial cerebrospinal fluid (ACSF) solution containing (in mM): 125 NaCl, 2.5 KCl, 2 CaCl$_2$, 1 MgCl$_2$, 1.25 NaH$_2$PO$_4$, 26 NaHCO$_3$, 20 glucose, and 0.5 L-glutamine at pH 7.4. Sulforhodamine-101 (SR101, 0.1 μM; Invitrogen) was added to the ACSF to reveal blood vessels and damaged GCL cells in the red fluorescence channel (*Euler et al., 2009*). The carboxygenated ACSF was constantly perfused through the recording chamber at 4 ml/min and kept at ~36°C throughout the entire experiment. After the dissection, retinae were bulk-electroporated with the synthetic fluorescent calcium indicator Oregon-Green 488 BAPTA-1 (OGB-1; hexapotassium salt; Life Technologies) (*Briggman and Euler, 2011*). To electroporate the GCL, the dissected retina was flat-mounted with the GCL facing up onto an Anodisc (#13, 0.1 μm pore size, 13 mm diameter, Cytiva), and then placed between two 4 mm horizontal platinum disk electrodes (CUY700P4E/L, Nepagene/Xceltis). The lower electrode was covered with 15 μl of ACSF, while a 10 μl drop of 5 mM OGB-1 dissolved in ACSF was suspended from the upper electrode and lowered onto the retina. Then, nine electrical pulses (~9.2 V, 100 ms pulse width, at 1 Hz) from a pulse generator/wide-band amplifier combination (TGP110 and WA301, Thurlby handar/Farnell) were applied and then, the electroporated retina on the Anodisc was transferred into the recording chamber, whereby the dorsal edge of the retina pointed away from the experimenter. The retina was left there for ~30 min to recover, as well as adapted to the light stimulation by displaying a binary dense noise stimulus (20×15 matrix, 40×40 μm² pixels, balanced random sequence) at 5 Hz before the recordings started.

### Two-photon Ca$^{2+}$ imaging

For the functional Ca$^{2+}$ imaging experiments, a MOM-type two-photon microscope (designed by W Denk, MPI, Heidelberg; purchased from Sutter Instruments/Science Products) (*Euler et al., 2019*; *Euler et al., 2009*) was employed. The microscope was equipped with a mode-locked Ti:Sapphire laser (MaiTai-HP DeepSee, Newport Spectra-Physics) tuned to 927 nm (ideal wavelength to excite OGB-1), two photomultiplier tubes serving as fluorescence detection channels for OGB-1 (HQ 510/84, AHF/Chroma) and SR101 (HQ 630/60, AHF), and a water immersion objective (CF175 LWD×16/0.8 W, DIC N2, Nikon, Germany). To acquire images, custom-made software (ScanM by M Müller and T Euler) running under IGOR Pro 6.3 for the operating system Microsoft Windows (WaveMetrics) was used and time-lapsed 64×64 pixel image scans (100×100 μm) at 7.8125 Hz were taken. Routinely, the optic nerve position and the scan field position were recorded to reconstruct their retinal positions. High-resolution images (512×512 pixel images) were recorded to support semi-automatic ROI detection.

## Light stimulation for two-photon Ca²⁺ imaging

For the light stimulation of the retinal tissue, a digital light processing (DLP) projector (lightcrafter [LCr], DPM-E4500UVBGMKII, EKB Technologies Ltd) was used to display the visual stimuli through the objective onto the retina, whereby the stimulus was focused on the photoreceptor layer (*Franke et al., 2019*). The LCr was equipped with a light-guide port to couple in external, band-pass filtered green and UV light-emitting diodes (LEDs; green: 576 BP 10, F37-576; UV: 387 BP 11, F39-387; both AHF/Chroma). The band-pass filter was used to optimize the spectral separation of mouse M- and S-opsins (390/576 Dualband, F59-003, AHF/Chroma). Both LEDs were synchronized with the scan retracing of the microscope. Stimulator intensity (as photoisomerization rate, $10^3$ P∗s⁻¹ per cone) was calibrated to range from ~0.5 (black image) to ~20 for M- and S-opsins, respectively. Steady illumination of ~$10^4$ P∗s⁻¹ per cone was present during the scan recordings due to the two-photon excitation of photopigments (*Euler et al., 2019*; *Euler et al., 2009*).

In total, three types of light stimuli were used for the imaging of Ca²⁺ in the GCL: (1) full-field chirp stimulus (700 µm ∅; see details here [*Baden et al., 2016*]), (2) bright moving bar (0.3×1 mm²) at 1 mm s⁻¹ in eight directions to probe direction and orientation selectivity, and (3) random binary noise with a checkerboard grid of 20×15 checks and a check size of 40 µm at 5 Hz for 5 min to map RFs. Light stimulus center and scan field center were aligned. Before each stimulus was presented, the baseline was recorded after the laser started scanning for at least 30 s to avoid immediate laser-induced effects on the retinal activity (*Euler et al., 2019*; *Euler et al., 2009*; *Szatko et al., 2020*).

## Animals and tissue preparation for electrophysiological recordings

Electrophysiological data were recorded from isolated retinae from four C57Bl/6J mice of 8–10 weeks. The experiment was performed in accordance with the institutional animal care standards of Sorbonne Université (Paris, France). The animals were housed in enriched cages with ad libitum food, and watering. The ambient temperature was between 22°C and 25°C, the humidity was between 50% and 70%, and the light cycle was 12–14 hr of light, 10–12 hr of darkness. After killing the animal, the eye was enucleated and transferred rapidly into oxygenated Ames medium (Sigma-Aldrich). Dissection was made under dim light condition as described previously (*Marre et al., 2012*; *Yger et al., 2018*).

## Electrophysiological recordings

For the electrophysiological recordings, we mounted a piece of retina onto a membrane and then lowered it with the ganglion cell side against a 252-channel MEA whose electrodes were spaced by 30 µm. During dissection and recordings, the tissue was perfused with oxygenated Ames solution and a peristaltic perfusion system with two independent pumps: PPS2 (Multichannel Systems GmbH). Mice retinae were kept at 35–37°C during the whole experiment. The data sampling rate was 20 kHz. The raw signal was acquired through MC Rack Multi-channel Systems software 4.6.2, it was high-pass filtered at 100 Hz, and the spikes were isolated using SpyKING CIRCUS 1.0.628 (*Yger et al., 2018*). Subsequent data analysis was done with custom-written Python codes. We extracted the activity of a total of 391 neurons. We kept cells with a low number of refractory period violations (<0.95%, with median <0.05% for all experiments, 2 ms refractory period) and whose template waveform could be well distinguished from the template waveforms of other cells. These constraints ensured a good quality of the reconstructed spike trains. In addition, we discarded neurons that showed no or almost no responses to chirp stimulus, preventing the correct cell classification.

## Light stimulation for electrophysiological recordings

A white-mounted LED (MCWHLP1, Thorlabs Inc) was used as a light source, and the stimuli were displayed using a Digital Mirror Device (DLP9500, Texas Instruments) and focused on the photo-receptors using standard optics and an inverted microscope (Nikon). The light level corresponded to photopic vision: 4.9×$10^4$ and 1.4×$10^5$ isomerizations/(photoreceptor.s) for S-cones and M-cones, respectively. In total, we displayed two stimuli: (1) a random binary checkerboard with check size of 42 µm for 30–40 min at 30 Hz, and (2) a full-field chirp stimulus as used for the two-photon Ca²⁺ imaging experiments. It was played at 50 Hz, containing 20 repetitions of 32 s length.

## Pharmacology

All used drugs were added to the carboxygenated, perfused ACSF solution 15 min prior to the second recording of the GCL scan fields. For the drugs, the respective concentrations were used (in µM): 100 (Z)-1-[N-(2-aminoethyl)-N-(2-ammonioethyl)amino]diazen-1-ium-1,2-diolate (DETA/NO) and 1 strychnine. The ACSF solution with and without drug application was always kept at ~36°C.

## Data analysis of two-photon Ca²⁺ imaging recordings

Image extraction and semi-automatic region-of-interest (ROI) detection were performed using Igor PRO 8. All analyses were organized and performed in a custom-written schema using DataJoint for Python (http://datajoint.github.io/; *Yatsenko et al., 2015*).

### Preprocessing

After the Ca²⁺ traces were extracted from individual ROIs, as described elsewhere (*Baden et al., 2016*; *Szatko et al., 2020*), the raw traces were detrended by subtracting a smoothed version $r_{smooth}$ of the trace from the raw one. Detrending was necessary to remove slow drifts in the signal that were unrelated to the light-induced response. The smoothed trace $r_{smooth}$ was computed by applying a Savitzky-Golay filter (*Press and Teukolsky, 1990*) of third polynomial order and a window length of 60 s using the Python SciPy implementation *scipy.signal.savgol_filter* (*Virtanen et al., 2020*).

$$r_{detrend} = r_{raw} - r_{smooth}$$

For the chirp and moving bar stimulus, detrended traces were averaged over repetitions; and in the case of the moving bar stimulus, reduced to the response average for the preferred motion direction of the cell (for details, see *Baden et al., 2016*). Finally, response averages were normalized by first subtracting the baseline activity (computed as the mean over the first second), and then by dividing by the maximum amplitude $max_t(|r(t)|) = 1$. This normalization was performed independently for each ROI, stimulus, and condition.

### Inclusion criterion

To include reliable cell responses for the performed analyses, two consecutive quality filtering steps were applied. At first, the response quality criterion, also termed quality index (QI), was computed for the moving bar ($QI_{MB} > 0.6$) and full-field chirp ($QI_{chirp} > 0.45$). Cells that passed either one of these two QIs in both recording conditions were included, otherwise they were discarded in the following analyses. As in *Baden et al., 2016*, the QI is defined as follows:

$$QI = \frac{Var[\langle C \rangle_r]_t}{\langle Var[C]_t \rangle_r}$$

where $C$ is the $T$ by $R$ response matrix (time samples by stimulus repetitions) and $\langle \rangle_x$ and $Var[]_x$ denote the mean and variance across the indicated dimension $x$, respectively. As a second step, cells were assigned to an RGC group using the RGC classifier, which returned the RGC group index and a confidence score (i.e. assignment probability to the predicted RGC group by the random forest classifier [*Qiu et al., 2022*]). Only cells that were assigned to one of the RGC groups (i.e. RGC index 1–32) were included, whereas cells assigned to a displaced AC group (i.e. RGC index 33–46) were rejected. Cells that exceeded the confidence score threshold of >0.25 were included.

### Suppression index

For each cell, the suppression index (SI) was measured by comparing the (absolute) negative area under the curve ($AUC_{neg}$) of the chirp and moving bar responses with the total area under the curve ($AUC_{neg}+AUC_{pos}$) of the entire response trace. For the absolute $AUC_{neg}$, the response was clipped for value <0.

$$SI = \frac{|AUC_{neg}|}{|AUC_{neg}| + |AUC_{pos}|}$$

## Trace difference Δ

To compute the trace difference between the sequentially recorded responses per cell, we subtracted the first response average to the chirp and moving bar stimuli, i.e., Ctrl 1, from the second recorded light-induced response, i.e., either NO or Ctrl 2. For the cell type-specific analysis, we computed the average trace differences per cell. For the feature-based analysis, we computed the average trace differences per feature and per cell type.

## On-Off index

The On-Off index was computed as

$$OOI = \frac{\langle r_{on} \rangle_t - \langle r_{off} \rangle_t}{\langle r_{on} \rangle_t + \langle r_{off} \rangle_t}$$

where $r_{on}$ and $r_{off}$ are defined as the separated time components of the moving bar response into its On- and Off-component. For each component, we computed the mean value of the discrete differences along the time axis clipped between 0 and 1 to estimate if there is a response to the particular feature.

## Classification of functional RGC types

For the functional classification of RGC types, we used a previously published RGC classifier (**Qiu et al., 2022**). The classifier, which uses a random forest classifier, was trained, validated, and tested on previously published RGC type responses (**Baden et al., 2016**). As input to the classifier, we used the responses to the standard set of stimuli, i.e., full-field chirp and moving bar, as well as soma sizes (separates alpha and non-alpha types) and the p-values of the permutation test for direction selectivity (separates DS and Non-DS types). For every cell, the RGC classifier outputs its type index and the confidence scores for all 46 types. The confidence score, as described in 'inclusion criterion', was used as a quality criterion.

## RF estimation

We mapped RFs of RGCs using the RF Python toolbox *RFEst* (**Huang et al., 2021**), following the procedure in **Baden et al., 2016**, with few modifications. The binary dense noise stimulus (20×15 matrix, (40 μm)$^2$ pixels, balanced random sequence; 5 Hz) was centered on the recording field. We computed the temporal gradients of the Ca$^{2+}$ signals from the detrended traces and clipped negative values:

$$\dot{c} = \max(0, \dot{r}_{detrend})$$

The stimulus $S(t)$ and the clipped temporal gradients $c$ were upsampled to 10 times the stimulus frequency to compute the gradient-triggered average stimulus:

$$\boldsymbol{F}(x, y, \tau) = \int_{t=0}^{T} \dot{c}(t) \boldsymbol{S}(x, y, t + \tau)$$

where $\boldsymbol{S}(x, y, t)$ is the stimulus, $\tau$ is the lag ranging from approximately −0.20 to 1.38 s, and $T$ is the duration of the stimulus. We smoothed these raw RFs using a 5×5 pixel and 1 pixel standard deviation Gaussian window for each lag. Then, we decomposed the RF into a temporal ($F_t(\tau)$) and spatial ($\boldsymbol{F}_s(x, y)$) component using singular value decomposition and scaled them such that $\max(|F_t|) = 1$ and $\max(|\boldsymbol{F}_s|) = \max(|\boldsymbol{F}|)$. RF quality was computed as:

$$QI_{\text{RF}} = 1 - \frac{Var[\boldsymbol{F}(x, y, \tau) - F_t(\tau)\boldsymbol{F}_s(x, y)]}{Var[\boldsymbol{F}(x, y, \tau)]}$$

Only RFs with $QI_{\text{RF}} > 0.45$ were used for the analysis.

For each sRF $\boldsymbol{F}_s$, we fit a 2D Difference of Gaussians using the Python package *astropy* (**Price-Whelan et al., 2018**). The mean and covariance matrices of the center and surround Gaussian fits were tied, except for a linear scaling of the covariance matrix. We defined the polarity $p \in \{-1, 1\}$

of the sRF as the sign of the model fit at its mean. Next, we computed the center RF of the sRF as $\boldsymbol{F}_s^c = \max(0, p \cdot \boldsymbol{F}_s)$. The surround index was computed as:

$$RF_{surround} = \frac{\sum_{x,y}(\boldsymbol{F}_s - \boldsymbol{F}_s^c)}{\sum_{x,y}(|\boldsymbol{F}_s|)}.$$

To measure the center RF size, we fit a 2D Gaussian to the center RF, with the mean fixed to the one obtained from the Difference of Gaussians fit. The area covered by two standard deviations of this Gaussian fit was used as the RF size.

## Functional clustering

The functional clustering was based on a similar approach as in *Baden et al., 2016*. The clustering was only applied on RGC types previously classified as $G_{32}$ and only recorded in Ctrl 1. First, visual features from the full-field chirp and moving bar $Ca^{2+}$ responses were extracted using sparse principal component analysis (PCA) (*Zou et al., 2006*). After optimizing the alpha parameter for each stimulus, each cell's dimensionality was reduced to 30 features, whereby the chirp covered 20 and the moving bar 10 features. Alpha was optimized in a way that every part of each stimulus was represented by one feature to increase interpretability. Each feature was standardized across cells before clustering. Then, the features were used to cluster the cells using a Mixture of Gaussian model. The ideal number of clusters was chosen based on the cross-validated BIC. Additionally, cluster coherence was computed and validated using intra- and inter-cluster correlation, as well as the influence of potential batch effects, i.e., a single cluster originates from a single retina or scan field, but is found across several ones. Then, the model was used to predict cell-type labels. Finally, cluster 3 showed a high signal-to-noise ratio, thus cells were re-clustered, which originated in three clusters, whereby two showed high variability in their chirp and moving bar responses. These cells were discarded in the further analysis to clean this cluster from potential contamination.

## Statistical analysis

To quantify the differences between traces, a Shaprio-Wilk test was used to test for normality and then either the two-sided t-test (if normally distributed) or the non-parametric Wilcoxon signed-rank test (Mann-Whitney U-test). To determine $\alpha$, Bonferroni correction was used, depending on the number of tests performed. To test the difference between traces against zero, we either used the t-test or the non-parametric Wilcoxon signed-rank test, depending on the distribution. A one-sampled t-test was performed to test the mean against a population mean of zero to quantify if the mean difference diverges from zero. For the statistical comparison of the suppression index between conditions per cluster, the non-parametric Kruskal-Wallis one-way analysis of variance and post hoc Dunnett's test and Bonferroni correction to determine the statistical significance between conditions were used.

## Data analysis of electrophysiological recordings

### Functional clustering

To cluster cells in different functional types, we based our analysis on the chirp and checkerboard stimulus responses and represented each RGC with a reduced vector. To obtain these vectors, first, we constructed PSTHs from the spikes evoked from the chirp stimulus, using a binning of 100 ms, and spike-triggered averages (STAs). Then, for each experiment, we z-scored all PSTHs and performed a PCA on them. We kept the number of components that were needed to explain 80% variance of the data (11–12 components for the PSTH). Second, we used the temporal profile (40 samples at 30 Hz) of each cell's STA obtained using the checkerboard stimulus. We z-scored it and performed a PCA, keeping two components, which explains around 80% of the variance. This adds information about the classical STA polarity of the RGCs. Third, we used the area of the ellipse fitted to the classical STA, as the product value of their major and minor axis $\sigma$ values. These areas were normalized from 0 to 1. In this way, we obtain a data vector of around 15 values, depending on the experiment, that describes each RGC according to their response to a chirp and a checkerboard. Then, we performed an agglomerative clustering, setting the threshold value in a way that all clusters looked homogeneous across PSTHs and STAs. This resulted in over-clustering producing between 12 and 44 RGC groups (from 29 to 149 cells in each experiment).

## RF estimation

To estimate sRFs and tRFs, we displayed a random binary checkerboard with check size of 42 µm for 30–40 min at 30 Hz. A three-dimensional (3D) STA (x, y, and time) was sampled using 40 time samples. The spatial STA presented across all the figures was obtained as the 2D spatial slice at the maximum value after smoothing. The temporal STA is the 1D time slice at that same value. A double Gaussian fit was performed on the resulting spatial STA, and the ellipse corresponding to a $2\sigma$ contour of the fit was plotted for all the figures.

## Pseudo-calcium transformation

In the last step, we assigned each cluster to one of the 32 types described in *Baden et al., 2016*. To do this, we used the $Ca^{2+}$-dataset to match it with the MEA-dataset. We based our analysis on their reported data in the Extended data from their Figure 1, where the authors link electrophysiology and $Ca^{2+}$ imaging by means of a convolution between a $Ca^{2+}$ event-triggered by a single spike.

We transformed the PSTHs by convolving them with a decaying exponential (see *Figure 7c*; *Goldin et al., 2022*), in which we adjusted the temporal decay constant to maximize the correlation of our clusters and theirs (median maximum correlation of 0.78). RGC types that presented strong responses to the modulating frequencies and amplitude were assigned correctly, while other types, which mostly respond to On/Off steps, were assigned in a second round of correlation match after excluding the former clusters. Besides the correlation of the chirp traces, we confirmed the correct assigning of clusters by checking that the ellipses of each type form a proper mosaic (see *Figure 7a and b*), that the spatial STAs look uniform, and the similarity of their spike waveforms. After doing this for every experiment, we grouped the cells of each type across animals where we found a clear homogeneity. Finally, we computed a correlation matrix between the average chirp response of each type to show that the clusters are homogeneous (see *Figure 7e*).

With this procedure, we were able to match a majority of RGC types between those datasets, yet aligning the datasets is challenging. In fact, the cell types underlying the $Ca^{2+}$ and MEA RGC clusters may not always be same. A caveat is that while $Ca^{2+}$ is a proxy for spiking activity, other $Ca^{2+}$ sources as well as sub-threshold membrane potential changes may affect the intracellular $Ca^{2+}$, potentially in a cell type-specific way.

## Functional cell typing of $G_{32}$

Once we obtained all the cells assigned to $G_{32}$, we pooled the data from the four experiments and performed the same steps as before, to further distinguish sub-groups as we did with the $Ca^{2+}$ imaging data. We first over-clustered again in this step and merged the similar sub-clusters in a last step. We obtained again, now in the electrophysiological data, three clearly differentiated subgroups. From an initial set of 46 cells, we obtained three subgroups of 17, 10, and 15 cells (*Figure 8c*), plus a non-homogeneous subgroup of four cells that did not fit into any of them and was discarded.

## Statistical analysis

To quantify the ellipses of the sRFs, we used the two-sides t-test. To quantify if the full width at half minimum (FWHM) values of the tRFs were significantly different, we performed repeated measures ANOVA and post hoc Dunnett's test. Only cells where we could compute sRFs or tRFs across the three conditions were used for these analyses.

## Acknowledgements

We thank Gordon Eske and Merle Harrer for technical support and Clint L Makino, Philipp Berens, and Robert Feil for fruitful discussions. This work was funded by the Deutsche Forschungsgemeinschaft (DFG, German Research Foundation) 335549539/GRK2381 and CRC 1233 (project number 276693517). M Goldin was funded by a fellowship from Fondation de France and by CNRS and by grant ANR PerBaCo. We acknowledge support from the Open Access Publication Fund of the University of Tübingen.

# Additional information

## Funding

| Funder | Grant reference number | Author |
| --- | --- | --- |
| Deutsche Forschungsgemeinschaft | 335549539 | Thomas Euler |
| Deutsche Forschungsgemeinschaft | CRC 1233 - 276693517 | Thomas Euler |
| Fondation de France | | Matías A Goldin<br>Olivier Marre |
| Centre National de la Recherche Scientifique | | Matías A Goldin<br>Olivier Marre |
| Agence Nationale de la Recherche | PerBaCo | Matías A Goldin<br>Olivier Marre |
| Deutsche Forschungsgemeinschaft | GRK2381 | Thomas Euler |

The funders had no role in study design, data collection and interpretation, or the decision to submit the work for publication.

## Author contributions

Dominic Gonschorek, Conceptualization, Data curation, Software, Formal analysis, Validation, Investigation, Visualization, Methodology, Writing - original draft, Project administration, Writing – review and editing; Matías A Goldin, Jonathan Oesterle, Software, Formal analysis, Validation, Methodology, Writing – review and editing; Tom Schwerd-Kleine, Data curation, Investigation, Methodology, Writing – review and editing; Ryan Arlinghaus, Data curation; Zhijian Zhao, Conceptualization, Methodology; Timm Schubert, Conceptualization, Resources, Writing – review and editing; Olivier Marre, Resources, Funding acquisition, Project administration, Writing – review and editing; Thomas Euler, Conceptualization, Resources, Supervision, Funding acquisition, Investigation, Writing - original draft, Project administration, Writing – review and editing

## Author ORCIDs

Dominic Gonschorek ⓘ https://orcid.org/0000-0001-5118-9817
Jonathan Oesterle ⓘ https://orcid.org/0000-0001-8919-1445
Zhijian Zhao ⓘ https://orcid.org/0000-0002-3302-1495
Thomas Euler ⓘ https://orcid.org/0000-0002-4567-6966

## Ethics

All animals for the two-photon Ca2+ imaging experiments were conducted at the University of Tübingen and were performed according to the laws governing animal experimentation issued by the German Government as well as approved by the institutional animal welfare committee of the University of Tübingen. For all experiments, we used retinae (n=26) from C57Bl/6J mice (n=14; JAX 000664) of either sex between the age of 4-16 weeks. All animals were kept in the local animal facility and housed under the standard 12h/12h day/night cycle at 22°C and a humidity of 55%. The following procedures were carried out under very dim red (> 650 nm) light. Before each imaging experiment, the animal was dark-adapted for >1h, then anesthetized with isoflurane (CP-Pharma) and sacrificed by cervical dislocation. Electrophysiological data were recorded from isolated retinae from four C57Bl/6J mice of 8-10 weeks. The experiments were performed in accordance with the institutional animal care standards of Sorbonne Université (Paris, France). The animals were housed in enriched cages with ad libitum food and watering. The ambient temperature was between 22 and 25°C, the humidity was between 50 and 70%, and the light cycle was 12-14h of light, 10-12h of darkness. After killing the animal, the eye was enucleated and transferred rapidly into oxygenated Ames medium (Sigma-Aldrich).

Reviewer #1 (Public review): https://doi.org/10.7554/eLife.98742.3.sa1
Reviewer #2 (Public review): https://doi.org/10.7554/eLife.98742.3.sa2

Author response https://doi.org/10.7554/eLife.98742.3.sa3

# Additional files

## Supplementary files
MDAR checklist

## Data availability
Data and custom analyses including code and notebooks to reproduce analyses is available at https://github.com/eulerlab/nitric_oxide_rgc_eLife (copy archived at *Gonschorek and Euler, 2025*) and https://doi.org/10.12751/g-node.mpf63w.

The following dataset was generated:

| Author(s) | Year | Dataset title | Dataset URL | Database and Identifier |
|---|---|---|---|---|
| Gonschorek D, Goldin MA, Oesterle J, Schwerd-Kleine T, Arlinghaus R, Zhao Z, Schubert T, Marre O, Euler T | 2024 | Dataset: Nitric oxide modulates contrast suppression in a subset of mouse retinal ganglion cells | https://doi.org/10.12751/g-node.mpf63w | German Neuroinformatics Node - G-Node, 10.12751/g-node.mpf63w |

The following previously published dataset was used:

| Author(s) | Year | Dataset title | Dataset URL | Database and Identifier |
|---|---|---|---|---|
| Baden T, Berens P, Franke K, Román Rosón M, Bethge M, Euler T | 2020 | Data from: The functional diversity of retinal ganglion cells in the mouse | https://doi.org/10.5061/dryad.d9v38 | Dryad Digital Repository, 10.5061/dryad.d9v38 |

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
