## [Editor Report · eLife Assessment]

This **important** study is the first comprehensive analysis of the modulatory effects of nitric oxide (NO) on the response properties of retinal ganglion cells (RGCs) in the mouse retina using two-photon calcium imaging and multi-electrode arrays (MEA). The results provide **compelling** evidence that a subset of suppressed-by-contrast RGCs are affected. These unexpected findings are likely of broad interest to visual neuroscientists.

---

## [Referee Report · Reviewer #1 (Public review)]

Summary:

Nitric oxide (NO) has been implicated as a neuromodulator in the retina. Specific types of amacrine cells (ACs) produce and release NO in a light-dependent manner. NO diffuses freely through the retina and can modulate intracellular levels of cGMP, or directly modify and modulate proteins via S-nitrosylation, leading to changes in gap-junction coupling, synaptic gain, and adaptation. Although these system-wide effects have been documented, it is not well understood how the physiological function of specific neuronal types is affected by NO. This study aims to address this gap in our knowledge.

There are two major findings. (1) About a third of the retinal ganglion cells display cell-type specific adaptation to prolonged stimulus protocols. (2) Application of NO specifically affected Off-suppressed ganglion cells designated as G32 cells. The G32 cluster likely contains 3 ganglion cell types that are differentially affected.

This is the first comprehensive analysis of the functional effects of NO on ganglion cells in the retina. The cell-type specificity of the effects is surprising and provides the field with valuable new information.

Strengths:

NO was expected to produce small effects, and considerable effort was expended in validating the system to ensure that changes in the state of the preparation would not confound any effects of NO. The authors used a sequential stimulus protocol to control for changes in the sensitivity of the retina during the extended recording periods. The approach potentially increases the sensitivity of the measurements and allows more subtle effects to be observed.

Neural activity was measured by Ca-imaging. Responsive ganglion cells were grouped into 32 types using a clustering analysis. Initial control experiments demonstrated that the cell-types revealed by the analysis largely recapitulate those from their earlier landmark study using a similar approach.

Application of NO to the retina modulated responses of a single cluster of cells, labeled G32, while having little effect on the remaining 31 clusters. In separate experiments, ganglion cell spiking activity was recorded on a multi-electrode array (MEA). Together the Ca-imaging and MEA recordings provide complementary approaches and demonstrate that NO modulates the temporal but not spatial properties of affected cell-types.

Weaknesses:

The concentration of NO used in these experiments was ~0.25µM, which is 5- to 10-fold lower than the endogenous concentration previously measured in rodent retina. It is perhaps surprising that this relatively low NO concentration produced significant effects. However, the endogenous measurements were done in an eye-cup preparation, while the current experiments were performed in a bare (no choroid) preparation. Perhaps the resting NO level is lower in this preparation. It is also possible that the low concentration of NO promoted more selective effects.

---

## [Referee Report · Reviewer #2 (Public review)]

Neuromodulators are important for circuit function, but their roles in the retinal circuitry are poorly understood. This study by Gonschorek and colleagues aims to determine the modulatory effect of nitric oxide on the response properties of retinal ganglion cells. The authors used two photon calcium imaging and multi-electrode arrays to classify and compare cell responses before and after applying a NO donor DETA-NO. The authors found that DETA-NO selectively increases activity in a subset of contrast-suppressed RGC types. In addition, the authors found cell-type specific changes in light response in the absence of pharmacological manipulation in their calcium imaging paradigm. This study focuses on an important question and the results are interesting. The limitations of the method and data interpretation are adequately discussed in the revised manuscript.

The authors have addressed my previous comments, included additional discussions on the limitations of the method, and provided a more careful interpretation of their data.

---

## [Author Response]

The following is the authors’ response to the current reviews.

**Reviewer #1 (Public review):**
Summary:Nitric oxide (NO) has been implicated as a neuromodulator in the retina. Specific types of amacrine cells (ACs) produce and release NO in a light-dependent manner. NO diffuses freely through the retina and can modulate intracellular levels of cGMP, or directly modify and modulate proteins via S-nitrosylation, leading to changes in gap-junction coupling, synaptic gain, and adaptation. Although these system-wide effects have been documented, it is not well understood how the physiological function of specific neuronal types is affected by NO. This study aims to address this gap in our knowledge.There are two major findings. (1) About a third of the retinal ganglion cells display cell-type specific adaptation to prolonged stimulus protocols. (2) Application of NO specifically affected Off-suppressed ganglion cells designated as G32 cells. The G32 cluster likely contains 3 ganglion cell types that are differentially affected.This is the first comprehensive analysis of the functional effects of NO on ganglion cells in the retina. The cell-type specificity of the effects is surprising and provides the field with valuable new information.Strengths:NO was expected to produce small effects, and considerable effort was expended in validating the system to ensure that changes in the state of the preparation would not confound any effects of NO. The authors used a sequential stimulus protocol to control for changes in the sensitivity of the retina during the extended recording periods. The approach potentially increases the sensitivity of the measurements and allows more subtle effects to be observed.Neural activity was measured by Ca-imaging. Responsive ganglion cells were grouped into 32 types using a clustering analysis. Initial control experiments demonstrated that the celltypes revealed by the analysis largely recapitulate those from their earlier landmark study using a similar approach.Application of NO to the retina modulated responses of a single cluster of cells, labeled G32, while having little effect on the remaining 31 clusters. In separate experiments, ganglion cell spiking activity was recorded on a multi-electrode array (MEA). Together the Ca-imaging and MEA recordings provide complementary approaches and demonstrate that NO modulates the temporal but not spatial properties of affected cell-types.Weaknesses:The concentration of NO used in these experiments was ~0.25µM, which is 5- to 10-fold lower than the endogenous concentration previously measured in rodent retina. It is perhaps surprising that this relatively low NO concentration produced significant effects. However, the endogenous measurements were done in an eye-cup preparation, while the current experiments were performed in a bare (no choroid) preparation. Perhaps the resting NO level is lower in this preparation. It is also possible that the low concentration of NO promoted more selective effects.
**Reviewer #2 (Public review):**
Neuromodulators are important for circuit function, but their roles in the retinal circuitry are poorly understood. This study by Gonschorek and colleagues aims to determine the modulatory effect of nitric oxide on the response properties of retinal ganglion cells. The authors used two photon calcium imaging and multi-electrode arrays to classify and compare cell responses before and after applying a NO donor DETA-NO. The authors found that DETA-NO selectively increases activity in a subset of contrast-suppressed RGC types. In addition, the authors found cell-type specific changes in light response in the absence of pharmacological manipulation in their calcium imaging paradigm. This study focuses on an important question and the results are interesting. The limitations of the method and data interpretation are adequately discussed in the revised manuscript.The authors have addressed my previous comments, included additional discussions on the limitations of the method, and provided a more careful interpretation of their data.
**Recommendations for the authors:**
Please correct the citation that reviewer #1 mentioned. In addition, a little more discussion of the NO concentration issue would be helpful. The low NO concentration is not a weakness in the data; it simply raises questions regarding the interpretation.

Thank you for these recommendations.

Regarding the citation error, we are not sure if Reviewer #1 refers to a citation formatting error or incorrect placement. In any case, we modified the text: We specified the extracted information regarding the NO concentrations and put the applied concentration into that context (Lines 621-635). In addition, we made clear that the citation of Guthrie (2014) refers to the dissertation, which can be easily retrieved via *Google Scholar*. We also cited the mentioned ARVO abstract by Guthrie and Mieler (2014).

We hope that these modifications solve the above-mentioned issues.

The following is the authors’ response to the original reviews.

**Reviewer #1 (Public Review):**
Summary:Nitric oxide (NO) has been implicated as a neuromodulator in the retina. Specific types of amacrine cells (ACs) produce and release NO in a light-dependent manner. NO diffuses freely through the retina and can modulate intracellular levels of cGMP, or directly modify and modulate proteins via S-nitrosylation, leading to changes in gap-junction coupling, synaptic gain, and adaptation. Although these system-wide effects have been documented, it is not well understood how the physiological function of specific neuronal types is affected by NO. This study aims to address this gap in our knowledge.Strengths:NO was expected to produce small effects, and considerable effort was expended in validating the system to ensure that any effects of NO would not be confounded by changes in the state of the preparation. The authors used a paired stimulus protocol to control for changes in the sensitivity of the retina during the extended recording periods. The approach potentially increases the sensitivity of the measurements and allows more subtle effects to be observed.Neural activity was initially measured by Ca-imaging. Responsive ganglion cells were grouped into 32 types using a clustering analysis. Initial control experiments demonstrated that the cell-types revealed here largely recapitulate those from their earlier landmark study using the same approach (Fig. 2).Application of NO to the retina strongly modulated responses of a single cluster of cells, labeled G32, while having little effect on the remaining 31 clusters. This result is evident in Fig. 3e.Separate experiments measured ganglion cell spiking activity on a multi-electrode array (MEA). Clustering analysis of the peri-stimulus spike-time histograms (PSTHs) obtained from the MEA data also revealed 32 clusters. The PSTHs for each cluster were aligned to the Ca-imaging data using a convolution approach. The higher temporal resolution of the MEA recordings indicated that NO increased the speed of sub-cluster 2 responses but had no effect on receptive field size. The physiological significance of the small change in kinetics remains unclear.

We thank the reviewer for their detailed and constructive comments.

Weaknesses:The G32 cluster was further divided into three sub-types using Bayesian Information Criterion (BIC) based on the temporal properties of the Ca-responses. This sub-clustering result seems questionable due to the small difference in the BIC parameter between 2 and 3 clusters. Three sub-clusters of the G32 cluster were also revealed for the PSTH data, however, the BIC analysis was not applied to further validate this result.

(1.1) We agree with the reviewer that this is an important point to be clarified. To this end, we repeated the analysis with n=2 clusters (see Author response image 1 below). In brief, we found that the overall interpretation did not change: Both clusters in the Ctrl1-dataset showed barely any type-specific adaptational effects, whereas under NO application, temporal contrast responses decreased (see Author response image 1 below). If requested, we would be happy to add this image to the supplementary material.

In an additional analysis, we evaluated if n=2 or n=3 was the “better” choice for the number of clusters. In the new Supplementary Fig. S4, we compared the clusters with n=2 (top) and n=3 (bottom). For n=2, the two clusters are relatively strongly correlated for both visual stimuli, whereas for n=3, the clusters become more distinct, especially with respect to differences in the correlations for the two stimuli (Fig. S4b). For n=2, the low intra-cluster correlation (ICC) strongly suggests that cluster 2 contains multiple response types (ICC(C2) = 0.5 ± 0.48, mean ± s.d.; Fig. S4c). For n=3, the mean ICC values are high for all three clusters (ICC(C1) = 0.81 ± 0.16; ICC(C2) = 0.86 ± 0.07; ICC(C3) = 0.83 ± 0.1; mean ± s.d.). Together, this suggests that n=3 clusters captures the response diversity in G32 better than n=2 clusters.

Finally, we performed a BIC analysis for the MEA dataset and found the optimal number of clusters to be also n=3 (see new Suppl. Fig. S5).

The alignment of sub-clusters 1, 2, and 3 identified in the Ca-imaging and the MEA recordings seemed questionable, because the temporal properties of clusters did not align well, nor did the effects of NO.

(1.2) To address this important point, we analyzed the correlations between the control responses of the three clusters from the Ca^2+^-dataset with the ones from the MEA-dataset (see new Suppl. Fig. S7). To avoid confusion, we named the clusters in the MEA-dataset i,ii,iii (see Fig. 8). We found two of the three clusters to be highly correlated (Ca^2+^ clusters 2,3 and MEA clusters iii, ii), whereas one cluster was much less so (cluster 1 vs. cluster i), likely due to differences in response kinetics. In clusters i and ii NO application led to a release of suppression for temporal contrasts – similar to what we observed in the Ca^2+^ data (see also our new analysis of the MEA data in Suppl. Fig. S6, as discussed further below).

We agree that the cell types underlying the Ca^2+^ and MEA G32 clusters may not be the same – aligning functional types between those two methods is challenging due to several factors, mainly because while Ca^2+^ is a proxy for spiking activity, other Ca^2+^ sources as well as sub-threshold membrane potential changes affect the intracellular Ca^2+^, potentially in a cell type-specific way. We explain this now better in the text.

In any case, our main point was not to unambiguously align the cell types but to show that in both datasets, we find three subclusters of G_32_, which are affected by NO in a differential manner, particularly their suppression to temporal contrasts.

The title of the paper indicates that nitric oxide modulates contrast suppression in a subset of mouse retinal ganglion cells, however, this result appears to be inferred from previous results showing that G32 is identified as a "suppressed-by-contrast" cell. The present study does not explicitly evaluate the amount of contrast-suppression in G32 cells.

(1.3) The reviewer is correct in that we did not quantify contrast-suppression in G_32_ in detail but focused on the responses to temporal contrast (chirp and moving bar) and its modulation by NO (Fig. 5). In this context, please note that G_32_’s responses to the moving bar stimulus suggests that the cells are also suppressed by spatial contrast (i.e., an edge appearing in their RF).The functional RGC type G_32_ (“Off suppressed 2”) was defined in an earlier study (Baden et al. 2016); it was assigned to the “Suppressed-by-Contrast” (SbC) category mainly because temporal contrast suppresses its responses. Already then, coverage analysis indicated that G_32_ may indeed contain several RGC types – in line with our clustering analysis. It is still unclear if G_32_ contains one (or more) of the SbC cells described by Jacoby & Schwartz (2018); in their recent study, Wienbar and Schwarz (2022) introduced the novel bursty-SbC RGC, which Goetz et al. (2022) speculated to potentially align with G_32_.

We now discuss the relationship between G_32_ and the SbC RGCs defined in other studies in the revised manuscript.

In its current form, the work is likely to have limited impact, since the morphological and functional properties of the affected sub-cluster remain unknown. The finding that there can be cell-specific adaptation effects during experiments on in vitro retina is important new information for the field.

(1.4) Again, we thank the reviewer for the detailed and helpful feedback. We hope that the reviewer finds our revised manuscript improved.

**Reviewer #1 (Recommendations For The Authors):**
Most of the calcium activity traces (dF/F) throughout the paper have neither vertical nor horizontal calibration bars. Presumably, most values are positive, but this is unclear as a zero level is not indicated anywhere. Without knowing where zero dF/F is, it is not possible to determine whether the NO increased the Ca-signal or blocked a decrease in the Ca-signal.

Both ∆F/F and z-scoring, as we used here, are ways to normalize Ca^2+^ traces. We decided against using ∆F/F_0_ because this typically assumes that F represents the cell’s Ca^2+^ resting level (F_0_; without activity). However, in our measurements, the “resting” Ca^2+^ levels (i.e. before presenting a stimulus) may indeed reflect no spiking activity (e.g., in an ON RGC) but may also reflect baseline spiking activity (e.g., in an G_32_, which has a baseline firing rate of ~10 Hz; see Fig. S6). Hence, we used z-scoring, which carries no assumption of resting Ca^2+^ level equal to no activity. In practice, we normalized all traces to the Ca^2+^ level prior to the light stimulus and defined this as zero (as described in the Methods).

We considered the reviewer’s suggestion of adding zero lines to every trace but felt that this would hamper the overall readability of the figures.

Regarding calibration bars: We made sure that horizontal bars (indicating time) are present in all figures. We decided to leave out vertical bars in Ca^2+^ responses, because as explained above, the traces are normalized (and unit-free), and within a figure all traces are scaled the same.

Points of clarification for the Methods:(1) The stimulus field was 800 x 600 µm. Presumably, both scan fields were contained within this region when scanning either Field 1 or Field 2 so that the adaptation level of the preparation at both locations was maintained?

Yes, the stimulation field is always kept centered on the respective recording (scan) field and the adaptation level for each recording field was maintained.

(2) There appeared to be an indeterminate amount of time between the initial 10-minute adaptation period and Ctrl1, whereas there were no such gaps between subsequent scans. Is this likely to produce differences in adaptation state and thus represent a systematic error?

At this time point, recording (scan) fields were selected to make sure that the cells in the field were uniformly labelled with the Ca^2+^ indicator and responsive to light stimuli. This typically happened already at the end of the light adaptation phase and/or right after. When selecting the fields, light stimuli were presented (to test responsiveness) and thereby the adaptation level was maintained independent of the duration of this procedure, minimizing systematic errors.

(3) Was the dense white noise stimulus applied during the wash-in period to maintain the adaptation state of the preparation prior to the subsequent scan?

The dense noise was not applied throughout the wash-in period but at least 5-10min before the field was recorded with a drug (e.g., NO).

Fig. 1d illustrates very nicely how the stimuli align with the responses. It would have been helpful to have this format continue throughout the paper but unfortunately, the vertical lines are dropped in Fig. 2a and then the stimulus waveform is omitted in Fig. 2e onwards.

Thanks, good idea. We added the vertical lines and the stimulus waveform to the figures where they were missing to improve the readability.

What was the rationale for selecting the concentration of the NO donor used? Is it likely to mimic natural levels?

A DETA/NO concentration of 100 µM is commonly used in studies investigating NOinduced effects. DETA/NO has a half-life time (t_0.5_) of 20 hours, which makes it more suitable for application in tissues (like our whole-mount preparation), because the donor can penetrate into the issue before releasing NO. In turn, this long t0.5 means that only a fraction of the bound NO is released per time unit.

Based on t_0.5_ for DETA/NO and NO, one can roughly estimate the NO range as follows: t_0.5_ of NO strongly depends on the tissue and is estimated in the second to minute range (Beckman & Koppenol, 1996). Assuming a t_0.5_ for NO of 2 minutes, a freshly prepared 100 µM DETA/NO solution is expected to result within the first hour a NO concentration of approx. 0.25 µM (taking into account that 1 mole of DETA/NO releases 1.5 moles of NO molecules; see Ramamurthi & Lewis 1997).

In general, it is difficult to determine the physiological concentration of NO in the retina. Different measurements point at peaks of a few 100 nM (e.g., frog retina, ganglion cells: 0.25 µM, Kalamkarov et al. 2016; rodent inner retina, 0.1 to 0.4 µM, Micah et al. 2014). Hence, the NO concentrations we apply should be within the measured physiological range.

Fig. 3e: what are the diamond symbols? If these are the individual cells, it might be better to plot them on top of the box plots so all are visible.

Indeed, the diamond symbols represent individual cells, yet outliers only. We decided not to plot all cells as a dot plot on top of the box plots since the readability will suffer as there are too many individual dots to show, e.g., n=251 for G_32_ Ctrl and n=135 for G_32_ DETA/NO.

Fig. 3: please explain more clearly the x-axis units in a-d and the y-axis units in e.

To estimate potential response differences between the first and the second scan (i.e. either Ctrl 2 or NO), the traces were subtracted cell-pairwise (∆ Ctrl: Ctrl 2 – Ctrl 1; ∆ DETA/NO: NO – Ctrl 1). As all Ca^2+^ traces were normalized, they are unit-free. Therefore, the x-axes in Fig. 3a-d represent the mean differences of each cell per cell type, e.g., a value of zero would mean that the traces of Ctrl 1 and Ctrl 2 for a cell are identical. The y-axis in Fig. 3e is also unit-free, because technically, it is the same measure as Fig. 3a-d. But since it summarizes the control- and NO-data, we refer to this as “delta mean trace.” We tried to make this clearer in the revised manuscript and a detailed description can be found in the Methods.

Fig. 3: "...a substantial number of RGC types (34%) changed their responses to chirp and/or moving bar stimuli in the absence of any pharmacological perturbation in a highly reproducible manner...". How many of the cell types showed a significant difference? Two cell-types with p<0.001are highlighted with 3 asterisks. It would be helpful to indicate on this plot which of the other cells showed significant differences.

Yes, this is a good idea. Thank you. We tried to add this information to the figure, but it became rather crowded. Therefore, we added a new Suppl. Fig. S3 (same style as Fig. 3) where we exclusively summarized the control-dataset.

Fig. 7: To illustrate the transform from PSTH to Ca-imaging, why not use G32 data as an example?

Fair point. We modified the figure and added G_32_ as an example.

It would be clearer if the cells were labeled consistently throughout the paper using their Baden cluster numbers rather than switching to the older nomenclature (JAM-B, local edge, alpha, etc), e.g. Fig. 7a,b.

In the revised manuscript, we now changed the nomenclature to the Ca2+ Baden et al. (2016) terminology. We used the alternative cell type names here because where Fig. 7a is discussed in the manuscript, the cell type matching did not happen yet. But we agree that a consistent nomenclature is helpful.

The evidence supporting the sub-clustering of the G32 cells for the two recording methods could have been stronger. In Fig. 5, the BIC difference between 2 and 3 clusters is rather small. Is this result robust enough to justify 3 rather than 2 clusters? The BIC analysis should also be performed on the PSTH data-set to support the notion that the MEA G32 cluster also contains 3 rather than 2 sub-clusters.

Regarding the sub-clustering of G_32_ into n=2 or n=3 clusters for both datasets, please see our detailed reply #1.1 in our response to the public comments above.

The alignment of the three sub-clusters across the Ca-imaging and MEA data looked questionable. For example, the cluster 2 and cluster 3 traces in Fig. 5e,f look similar, with cluster 1 being more different. In Fig. 8c on the other hand, cluster 1 and 3 look similar with cluster 2 being more different. The pharmacological results also did not align well. For the Ca-imaging, NO appeared to have a large effect on cluster 1, a more modest effect on cluster 2 and less effect on cluster 3 (Fig. 5f). In comparison, the MEA results diverged, with NO producing the largest effect on cluster 2 and very modest if any effects on clusters 1 and 3 (Fig. 8c). Moreover, the temporal properties of cluster 1 and cluster 3 look very different between the Ca-imaging and MEA data. Without further comment, these differences raise concerns about the reliability of the clustering and the validity of comparisons made across the two sets of experiments.

We agree that this is a critical point. Please see our reply #1.2 in our response to the public comments above.

Fig. 8: Transforming the PSTHs into Ca-traces is important to align the MEA recordings with the Ca-imaging data. It would also be very informative to see a more detailed overall presentation of the PSTH data since it provides a much higher temporal resolution of the responses. For example, illustrating the average PSTHs for the G32 cells under all the experimental conditions could be quite illuminating.

To address this point, we added a new Supplementary Fig. S6, which shows the pseudo-Ca^2+^ traces for each cluster and condition next to the PSTHs. In addition, we quantified the cumulative firing rate for response features (time windows) where temporal suppression was observed in the Ca^2+^ data. This new analysis shows that during NO-application, we can see an increase in firing rate in all clusters. Nevertheless, the effect of NO on the PSTHs is admittedly small and it is better visible in the pseudo-Ca^2+^ transformed traces. One possible explanation for this difference may be that the overall firing rates are quite dynamic in G_32_ such that a significant increase in “suppression” phases relative to the peak firing appears small.

**Reviewer #2 (Public Review):**
Neuromodulators are important for circuit function, but their roles in the retinal circuitry are poorly understood. This study by Gonschorek and colleagues aims to determine the modulatory effect of nitric oxide on the response properties of retinal ganglion cells. The authors used two photon calcium imaging and multi-electrode arrays to classify and compare cell responses before and after applying a NO donor DETA-NO. The authors found that DETA-NO selectively increases activity in a subset of contrast-suppressed RGC types.In addition, the authors found cell-type specific changes in light response in the absence of pharmacological manipulation in their calcium imaging paradigm. While this study focuses on an important question and the results are interesting, the following issues need further clarification for better interpretation of the data.

We thank the reviewer for her/his detailed and constructive comments.

(1) Design of the calcium imaging experiments: the control-control pair has a different time course from the control-drug pair (Fig 1e). First, the control-control pair has a 10 minute interval while the control-drug pair has a 25 minute interval. Second, Control 1 Field 2 was imaged 10 min later than Control 1 Field 1 since the start of the calcium imaging paradigm.Given that the control dataset is used to control for time-dependent adaptational changes throughout the experiment, I wonder why the authors did not use the same absolute starting time of imaging and the same interval between the first and second round of imaging for both the control-control and the control-drug pairs. This can be readily done in one of the two ways: 1. In a set of experiment, add DETA/NO between "Control 1 Field 1 and "Control 2 Field 1" in Fig. 1e as the drug group; or 2. Omit DETA/NO in the Fig. 1e protocol as the control group to monitor the time course of adaptational changes.

Thank you for raising this point. We hope that in the following we can clarify the reasoning behind our protocol and the analysis approach.

(2.1) Initially, we performed these experiments in different ways (also in the sequence suggested by the reviewer), before homing in on the paradigm illustrated in Fig. 1. We chose this paradigm for two reasons: First, we wanted to have for each retina both Ctrl1/Ctrl2 and Ctr1/NO data sets, to be sure that the time-dependent (adaptational) effects were not related to the general condition of an individual retina preparation. Second, we did not see obvious differences in time-dependent or NO-induced effects between paradigms. Therefore, while we cannot exclude that the absolute time between recordings can affect the observed changes, we do not think that such effects are substantial enough to change our conclusions.

In the revised manuscript, we now explicitly point at the different intervals.

Related to the concern above, to determine NO-specific effect, the authors used the criterion that "the response changes observed for control (ΔR(Ctrl2−Ctrl1)) and NO (ΔR(NO−Ctrl1)) were significantly different". This criterion assumes that without DETA-NO, imaging data obtained at the time points of "Control 1 Field 2" and "DETA/NO Field 2" would give the same value of ΔR as ΔR(Ctrl2−Ctrl1) for all RGC types. It is not obvious to me why this should be the case, because of the unknown time-dependent trajectory of the adaptational change for each RGC type. For example, a RGC type could show stable response in the first 30 min and then change significantly in the following 30 min. DETA/NO may counteract this adaptational change, leading to the same ΔR as the control condition (false negative). Alternatively, DETA/NO may have no effect, but the nonlinear timedependent response drift can give false positive results.

(2.2) Initially, we assumed that after adapting the retina to a certain light level, RGCs exhibit stable responses over time, such that when adding a pharmacological agent, we can identify drug-induced response changes (e.g., by calculating the response difference). To our surprise, we found that for some RGC types the responses changed between the first and the second recording (referred to as *cell type-specific adaptational effects*), which is why we devised the Ctrl1/Ctrl2 vs. Ctr2/NO analysis.

The reviewer is correct in that we assume in our analysis that the adaptational- and NO-induced effects are independent and sum linearly. Further, we agree with the reviewer that there may be other possibilities, two of which are highlighted by the reviewer:

(a) Interaction: for instance, if NO compensates for the adaptational effect, we would not be able to measure this; or, if this compensation was partial, underestimate both effects.

(b) More complex time-dependency: for example, if an RGC shows a pronounced adaptational effect with a longer delay (i.e. only after the second scan), or that a very transient NO effect has already disappeared when we perform the second scan. On the one hand, as we only can take snapshots of the RGC responses, we cannot exclude these possibilities. On the other hand, both effects (adaptational- and NO-dependent) were type-specific and reproducible between experiments (also with varying timing, see reply #2.1), which makes complex time dependencies less likely.

The revised manuscript now reflects these limitations of our recording paradigm and points out which effects can be detected, and which likely not.

I also wonder why washing-out, a standard protocol for pharmacological experiments, was not done for the calcium protocol since it was done in the MEA experiments. A reversible effect by washing in and out DETA/NO in the calcium protocol would provide a much stronger support that the observed NO modulation is due to NO and not to other adaptive changes.

(2.3) We agree that a clear wash-out would strengthen our findings. Indeed, in the beginning of our experiments, we tried to wash-out the agent in the Ca^2+^ recordings, as we did in the MEA recordings. We soon stopped doing this in the Ca^2+^ experiments, because response quality decreased for the third scan of the same field, likely due to bleaching of fluorescent indicator and photopigment. This is why we typically restrict the total recording time of the same field of RGCs to about 30 min (~ two scans with all light stimuli). Moreover, our MEA data showed that DETA/NO can largely be washed-out, which supports that we observed NO-specific effects. Therefore, we decided against further attempts to establish the wash-out also in the Ca^2+^ experiments (e.g., shortening the recording time by presenting fewer light stimuli).

(2) Effects of Strychnine: In lines 215-219, " In the light-adapted retina, On-cone BCs boost light-Off responses in Off-cone BCs through cross-over inhibition (83, 84) and hence, strychnine affects Off-response components in RGCs - in line with our observations (Fig. S2)" However, Fig. S2 doesn't seem to show a difference in the Off-response components. Rather, the On response is enhanced with strychnine. In addition, suppressed-by-contrast cells are known to receive glycinergic inhibition from VGluT3 amacrine cells (Tien et al., 2016). However, the G32 cluster in Fig. S2 doesn't seem to show a change with strychnine. More explanation on these discrepancies will be helpful.

(2.4) We thank the reviewer for this comment. Regarding the first part, we agree that the figure does not support differences in the Off-response components. We therefore rephrased the corresponding text accordingly. Additionally, we now show all RGC types with n>3 cells per recording condition in the revised Suppl. Fig. S2 and added statistics.

Regarding the second part, there are several possible explanations for these discrepancies:

(a) The SbC (transient Off SbC) studied in Tien et al. (2016) likely corresponds to the RGC type G_28_ (see Höfling et al. 2024). As mentioned above (see reply #1.2), it is unclear if G_32_ corresponds to a previously described SbC, and if so, to which. Goetz et al. (2022) proposed that G_32_ may align with the bursty-SbC (bSbC) type (their Supplemental Table 3), as described also by Wienbar and Schwartz (2022). An important feature of the bSbC type is that its contrast response function is mainly driven by intrinsic properties rather than synaptic input. If G_32_ indeed included the bSbC, this may explain why strychnine does not interfere with the suppression of temporal contrast.

(b) In Tien et al. (2016), the authors genetically removed the VG3-ACs (see their Fig. 3) and show that this ablation reduces the inhibition of tSbC cells in a stimulus size-dependent manner. Specifically, larger light stimuli (600 µm) only show marginal effects on the IPSCs and inhibitory synaptic conductance (see their Figs. 3c,d and 3e,f, respectively). In our study, the full-field chirp had a size of 800 x 600 µm. Therefore – and assuming that G_32_ indeed included tSbCs – our observation that strychnine did not affect temporal suppression in the full-field chirp responses would be in line with Tien et al. (2016).

(3) This study uses DETA-NO as an NO donor for enhancing NO release. However, a previous study by Thompson et al., Br J Pharmacol. 2009 reported that DETA-NO can rapidly and reversible induce a cation current independent of NO release at the 100 uM used in the current study, which could potentially cause the observed effect in G32 cluster such as reduced contrast suppression and increased activity. This potential caveat should at least be discussed, and ideally excluded by showing the absence of DETA-NO effects in nNOS knockout mice, and/or by using another pharmacological reagent such as the NO donor SNAP or the nNOS inhibitor l-NAME.

Thank you for pointing out this potential caveat. We certainly cannot exclude such side effects. However, we think that this explanation of our observations is unlikely, because Thompson et al. barely see effects at 100 µM DETA/NO; in fact, their data suggests that clear NO-independent effects on the cation-selective channel occur at much higher DETA/NO concentrations, such as 3 mM.

In any case, in the revised manuscript, we refer to this paper in the *Discussion*.

(4) Clarification of methods: In the Methods, lines 1119-1127, the authors describe the detrending, baseline subtraction, and averaging. Then, line 1129, " the mean activity r(t) was computed and then traces were normalized such that: max t(|r(t)|) = 1. How is the normalization done? Is it over the entire recording (control and wash in) for each ROI? Or is it normalized based on the mean trace under each imaging session (i.e. twice for each imaging field)?

The normalization (z-scoring) was done for each ROI individually per stimulus and condition (Ctrl 1, Ctrl 2, DETA/NO). We normalized the traces, because the absolute Ca^2+^ signal depends on factors, such as “resting” state of the cell (e.g., silent vs. baseline spiking activity in the absence of a light stimulus) and its fluorescent dye concentration. This also means that absolute response amplitudes are difficult to interpret. Hence, we focused on analyzing relative changes per ROI and condition, which still allowed us to investigate adaptational and drug-induced effects. In the revised manuscript, we changed the corresponding paragraph for clarification.

As for the clustering of RGC types, I assume that each ROI's cluster identity remains unchanged through the comparison. If so, it may be helpful to emphasize this in the text.

Yes, this is correct. We identified G_32_ RGCs based on their Ctrl1 responses and then compares these responses with those for Ctrl2 or NO. We now clarified this in the revised manuscript.

**Reviewer #2 (Recommendations For The Authors):**
The manuscript would benefit from a discussion of how the findings in this study relate to known mechanisms of NO modulation and previously reported effects of NO manipulations on RGC activity.

Thank you for the recommendation. We already refer to known mechanisms of NO within the retina in the Introduction. In the revised manuscript, we now added information to the Discussion.

In the abstract, "a paired-recording paradigm" could be misleading because paired recording generally refers to the simultaneous recording of two neurons. However, the paradigm in this study is essentially imaging experiments done at two time points.

We agree with the reviewer. To avoid any confusion with paired electrophysiological recordings, we changed the term “*paired-recording paradigm”* to “*sequential recording paradigm”* and replaced the term “*pair-/ed”* with “*sequentially recorded”*.